# DJ-1 suppresses ferroptosis through preserving the activity of S-adenosyl homocysteine hydrolase

Ji Cao[1,2], Xiaobing Chen[1,2], Li Jiang[1], Bin Lu[1], Meng Yuan[1], Difeng Zhu[1], Hong Zhu[1], Qiaojun He[1], Bo Yang[1] & Meidan Ying[1✉]

Ferroptosis is a newly characterized form of regulated cell death mediated by iron-dependent accumulation of lipid reactive oxygen species and holds great potential for cancer therapy. However, the molecular mechanisms underlying ferroptosis remain largely elusive. In this study, we define an integrative role of DJ-1 in ferroptosis. Inhibition of DJ-1 potently enhances the sensitivity of tumor cells to ferroptosis inducers both in vitro and in vivo. Metabolic analysis and metabolite rescue assay reveal that DJ-1 depletion inhibits the transsulfuration pathway by disrupting the formation of the S-adenosyl homocysteine hydrolase tetramer and impairing its activity. Consequently, more ferroptosis is induced when homocysteine generation is decreased, which might be the only source of glutathione biosynthesis when cystine uptake is blocked. Thus, our findings show that DJ-1 determines the response of cancer cells to ferroptosis, and highlight a candidate therapeutic target to potentially improve the effect of ferroptosis-based antitumor therapy.

---

[1] Institute of Pharmacology and Toxicology, Zhejiang Province Key Laboratory of Anti-Cancer Drug Research, College of Pharmaceutical Sciences, Zhejiang University, Hangzhou, China. [2] These authors contributed equally: Ji Cao, Xiaobing Chen. ✉email: mying@zju.edu.cn

Ferroptosis, a newly identified form of regulated cell death, is characterized by iron and lipid reactive oxygen species (ROS) accumulation and smaller mitochondria with condensed mitochondrial membrane densities, but does not share morphological, biochemical, or genetic similarities with other forms of regulated cell death, such as apoptosis[1]. Increasing evidence suggests that ferroptosis dysfunction is highly related to various human diseases, including tumorigenesis[2,3]. Therefore, multiple inducers of ferroptosis hold great potential for cancer therapy[4–8]. As such, there has been increased interest in the identification of the molecular components regulating ferroptosis. In addition to the cystine/glutamate transporter (SLC7A11, xCT)[1,9] and phospholipid hydroperoxide glutathione peroxidase (GPX4)[7], several proteins, such as acyl-CoA synthetase long-chain family member 4 (ref. [10]) and nuclear factor erythroid 2-related factor 2 (NRF2)[8], have been discovered to regulate ferroptosis over the past 7 years. However, the precise regulation of ferroptosis in specific disease contexts remains largely unknown, and little progress has been achieved in utilizing and studying the implication of ferroptosis-based treatment for cancer.

It has been well documented that DJ-1 plays a critical role in antioxidative stress[11], which might be essential for tumorigenesis and cancer progression[12]. To protect cells from oxidative stress-triggered cell death, DJ-1 directly eliminates ROS by oxidizing itself at the Cys106 residue to $-SO_2$ or $-SO_3$ (ref. [13]). Notably, DJ-1 is thought to stabilize the antioxidant transcriptional master regulator NRF2 (ref. [11]), which orchestrates the expression of genes coding for stress response and antioxidant proteins. In addition, we have shown that the different oxidation states of DJ-1 function as a cellular redox sensor determining the cell fate by either activating autophagy or apoptosis by regulating the activity of ASK1 (ref. [14]). While the antioxidative stress effects of DJ-1 are clear, all of the conclusions drawn so far are based on the activity against soluble ROS (such as $H_2O_2$)[15–17]. To date, there is no experimental evidence to show the effect of DJ-1 on lipid ROS, whose properties are completely different from those of soluble ROS, and specifically trigger ferroptotic cell death. Thus, it is interesting to ask whether DJ-1 plays a role in regulating ferroptosis.

In this study, we demonstrate that DJ-1 plays a pivotal role in cytoprotection against ferroptosis by maintaining cysteine synthesized from the transsulfuration pathway, which is the predominant source for biosynthesizing reduced glutathione (GSH) after blocking xCT-mediated cystine import. Metabolic analysis and metabolite rescue assays further reveal that S-adenosyl homocysteine hydrolase (SAHH), the only known enzyme that catalyzes the hydrolysis of SAH to homocysteine in the transsulfuration pathway, is involved in this process. Specifically, DJ-1 determines the formation of the SAHH tetramer and its enzymatic activity by altering the interaction between SAHH and its negative regulator AHCYL1. Moreover, suppression of the DJ-1 expression markedly increases the sensitivity of tumor cells to the ferroptosis inducer erastin both in vitro and in vivo. The discovery of DJ-1 as a negative modular of ferroptosis provides opportunities to facilitate ferroptosis-based cancer therapy.

## Results

**DJ-1 is a bona fide negative ferroptosis modulator**. To investigate the effect of DJ-1 on ferroptosis, we established several stable DJ-1 knockdown (KD) cancer cell lines (H1299, PANC1, A549, H292, and H838) by using two specific short hairpin RNA (shRNAs) against DJ-1 (referred to as DJ-1 KD #1 and #2; Fig. 1a, Supplementary Fig. 1a). Suppression of DJ-1 expression by RNA interference significantly promoted erastin-induced accumulation of lipid ROS, a classic biomarker of ferroptosis[1], in all five of these

cell lines as determined by flow cytometry using the fluorescent probe C11-BODIPY (Fig. 1b, Supplementary Fig. 1b). Similar results were observed using three other ferroptosis inducers, sorafenib[8], RSL3 (ref. [7]), and ML210 (ref. [18]) in H1299 cells (Supplementary Fig. 1c), suggesting that the loss of DJ-1 enhances ferroptosis. We next monitored the morphological changes of mitochondria, another ferroptotic event, through transmission electron microscopy. In line with the lipid ROS results, we observed shrunken mitochondria with increased membrane density in erastin-treated cells, and this effect was dramatically enhanced by DJ-1 KD (Fig. 1c). Moreover, the effect of erastin-triggered ferroptotic cell death induction was dramatically enhanced in DJ-1 KD cells (Fig. 1d). Additionally, the activation of p38 mitogen-activated protein kinase (MAPK) is involved in erastin-induced ferroptosis[19]. Our results indicated that DJ-1 KD significantly increased the phosphorylation levels of p38 induced by erastin (Supplementary Fig. 1d). To validate that DJ-1 KD specifically enhances the ferroptotic process, we used ferrostatin-1 (Fer-1), a specific inhibitor of ferroptosis[1,20]. Notably, Fer-1 treatment completely reversed erastin-induced lipid ROS accumulation and cell death both in control and DJ-1 KD cells (Fig. 1e, f, Supplementary Fig. 1e). Thus, under ferroptotic stimuli, KD of DJ-1 markedly amplified the ferroptosis-inducing effect. To further strengthen our conclusion, we generated three DJ-1 knockout (KO) subclones from H1299 cells with two single guide RNAs through CRISPR/Cas9-based KO technology (Supplementary Fig. 1f, g). As expected, the three DJ-1 KO subclones exhibited more erastin-induced accumulation of lipid ROS and more cell death compared to those of the three wild-type subclones (Fig. 1g, h).

Next, we asked whether ectopic expression of DJ-1 protein in cells could diminish erastin-induced ferroptosis. Although the ectopic DJ-1 expression level was approximately two to five fold higher than that of the control group, we didn't observe any changes in lipid ROS accumulation or cell death caused by erastin in four ferroptosis-sensitive cancer cell lines (H1299, A2780, 786-O, and KHOS; Supplementary Fig. 2a–e). However, when we reintroduced human DJ-1 into DJ-1$^{-/-}$ mouse embryonic fibroblast cells (MEFs; Fig. 2a), overexpression of DJ-1 significantly reduced erastin-induced ferroptotic events, including lipid ROS accumulation (Fig. 2b), cell death (Fig. 2c), and viability (Fig. 2d). Moreover, similar effects were also observed in cells treated with sorafenib (Supplementary Fig. 2f). These data suggest that the protective effect of overexpressed DJ-1 against ferroptosis is not a linear response, especially in cells with higher basal levels of DJ-1, which has been extensively observed in many cancer cell lines.

While mutations in DJ-1 (PARK7) are a known cause of early-onset autosomal recessive Parkinson's disease[21], these mutations have also been found in tumor tissues (http://cancer.sanger.ac.uk/cosmic). Thus, we thought it would be interesting to ask whether these mutants of DJ-1 would exhibit the same effect on ferroptosis. As shown in Fig. 2e, f, only wild-type DJ-1, but not the disease-associated DJ-1 mutants (M26I, E64D, R98Q, A104T, D149A, G150S, E163K, L166P, and A171S), decreased the ferroptotic cell death induced by erastin. This suggests that the proper functioning of DJ-1 is required for its ferroptosis-blocking effect. Collectively, these data demonstrate that DJ-1 is a bona fide negative modulator regulating ferroptosis.

**NRF2 is not involved in DJ-1-regulated ferroptosis**. We next investigated the mechanism of DJ-1-regulated ferroptosis. Several lines of evidence suggest that DJ-1 stabilizes the antioxidant transcriptional master regulator NRF2 by preventing its association with its inhibitor protein, KEAP1 (ref. [11]). Recently, it was reported

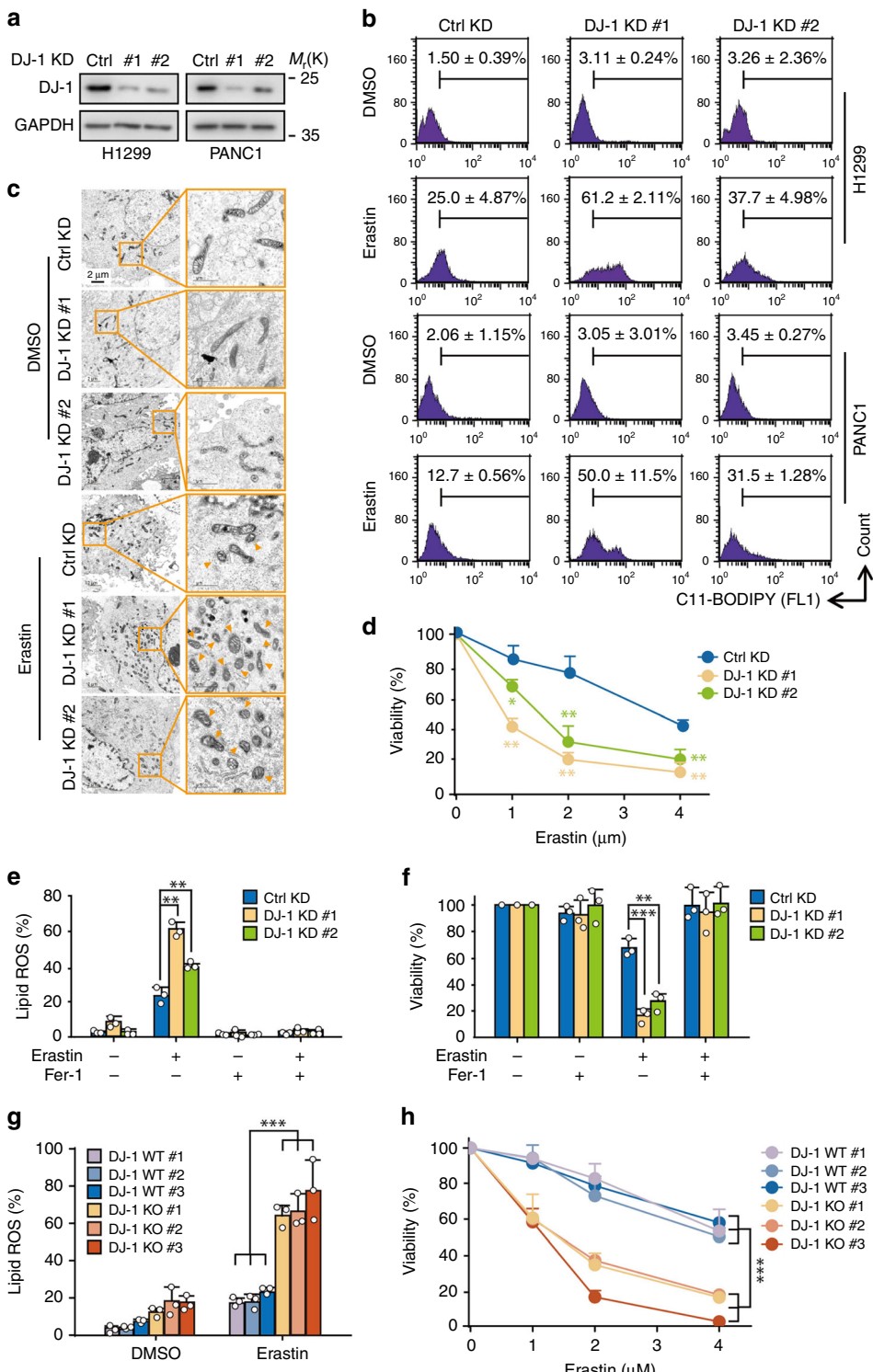

**Fig. 1 DJ-1 KD enhances ferroptotic cell death. a** Western blot analysis of DJ-1 expression in indicated DJ-1 KD H1299 and PANC1 cells. Independent experiments are repeated three times and representative data are shown. **b** Indicated KD H1299 and PANC1 cells were treated with erastin (2 μM) for 12 h, and lipid ROS production was assayed by flow cytometry using C11-BODIPY. Representative data are shown and the statistical analysis is from three independent experiments. **c** Representative transmission electron microscopy images of control KD or DJ-1 KD H1299 cells treated with erastin (2 μM, 12 h). A minimum of ten cells per treatment condition were examined. Orange triangle representative's shrunken mitochondria. Independent experiments are repeated three times and representative data are shown. **d** Indicated DJ-1 KD H1299 cells were treated with erastin (1–4 μM) for 36 h, and cell viability was assayed. **e** Indicated DJ-1 KD H1299 cells were treated with erastin (2 μM) with or without ferrostatin-1 (Fer-1, 1 μM) for 12 h, and lipid ROS production was assayed. **f** Cell viability was assayed in indicated DJ-1 KD H1299 cells treated for 36 h with erastin (2 μM) with or without Fer-1 (1 μM). **g** DJ-1 WT and KO H1299 cells were treated with erastin (2 μM) for 12 h, and lipid ROS production was assayed. **h** Cell viability was assayed in indicated DJ-1 WT and KO H1299 cells treated for 36 h with erastin (1–4 μM). Data shown represent mean ± SD from three independent experiments. Comparisons were made using the two-tailed, unpaired Student's $t$-test; *$p < 0.05$, **$p < 0.01$, ***$p < 0.001$.

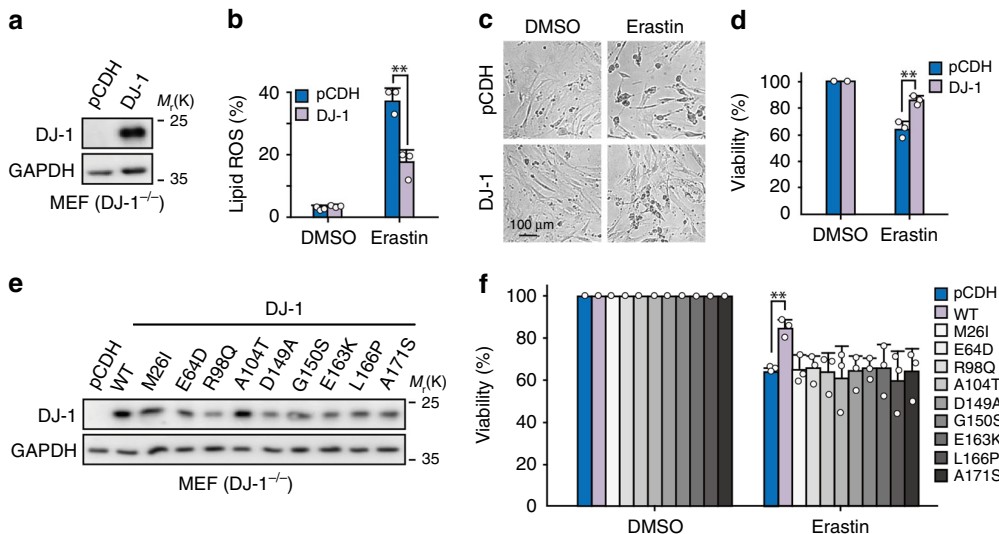

**Fig. 2 DJ-1 negatively regulates ferroptosis. a** Western blot analysis of DJ-1 expression in DJ-1$^{-/-}$ MEFs with reintroducing human DJ-1 protein. Independent experiments are repeated three times and representative data are shown. **b** Indicated MEFs were treated with erastin (400 nM) for 12 h, and lipid ROS production was assayed by flow cytometry using C11-BODIPY. **c** Phase-contrast images of MEFs treated with 400 nM erastin for 24 h. Independent experiments were repeated three times and representative data are shown. **d** Cell viability was assayed in indicated cells treated with 400 nM erastin for 24 h. **e** Western blot analysis of DJ-1 in DJ-1$^{-/-}$ MEFs infected with wild type or mutants of DJ-1. Independent experiments are repeated three times and representative data are shown. **f** Indicated MEFs were treated with erastin (400 nM) for 24 h, and cell viability was assayed. Data shown represent mean ± SD from three independent experiments. Comparisons were made using the two-tailed, unpaired Student's t-test; **$p < 0.01$.

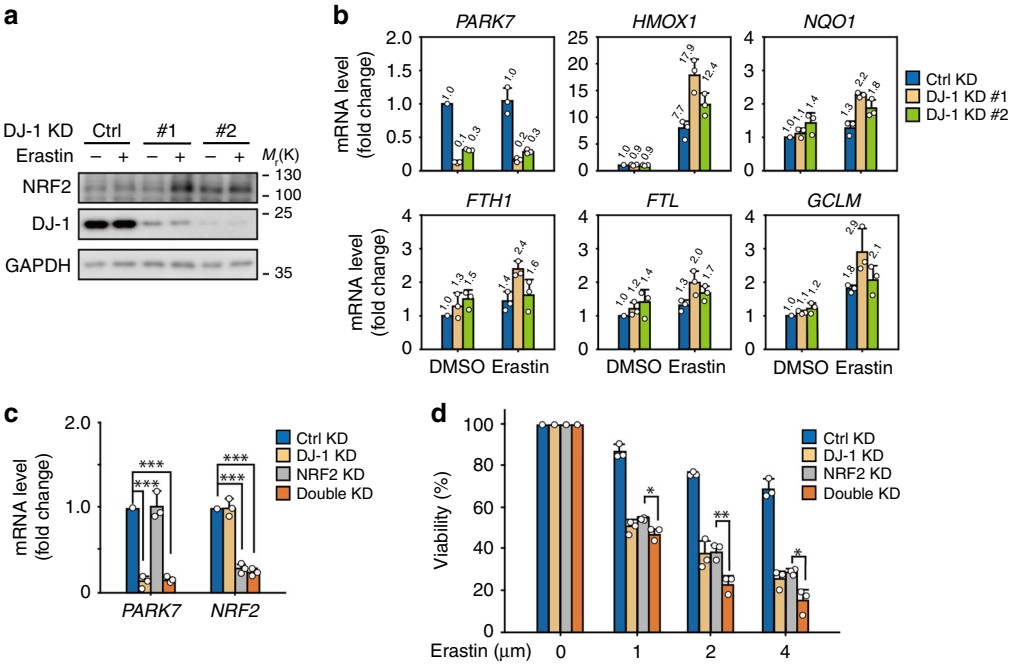

**Fig. 3 NRF2 is not involved in DJ-1-regulated ferroptosis. a, b** Indicated DJ-1 KD H1299 cells were treated with erastin (2 μM) for 12 h. Independent experiments are repeated three times and representative data are shown. **a** The protein expression of NRF2, and **b** its known downstream genes were assayed. The relative gene expression is normalized to β-actin. **c** The mRNA expression of PARK7 and NRF2 in indicated H1299 cells was assayed by qRT-PCR (#1 sequence of DJ-1 KD was used here). The relative gene expression is normalized to β-actin and the error bar indicates the SD value from triplicates. **d** Viability of H1299 cells infected with shRNAs for 72 h (#1 sequence of DJ-1 KD was used here) and treated with erastin (1–4 μM, 24 h). Data shown represent mean ± SD from three independent experiments. Comparisons were made using the two-tailed, unpaired Student's t-test; *$p < 0.05$, **$p < 0.01$, ***$p < 0.001$.

that NRF2 can negatively regulate ferroptosis[8]. Based on these findings, we first examined the protein levels of NRF2 under erastin treatment. To our surprise, KD of DJ-1 significantly upregulated the expression of NRF2 protein after erastin treatment in H1299 cells (Fig. 3a). Consistent with this finding, expression of well-characterized NRF2 target genes involved in regulating ferroptosis, including heme oxygenase 1 (HMOX1), NAD(P)H quinone dehydrogenase 1 (NQO1), ferritin heavy chain 1 (FTH1),

ferritin light chain (*FTL*), and glutamate–cysteine ligase modifier subunit (*GCLM*), did not decrease in DJ-1 KD cells (compared to that of control cells) under both basal and erastin treatment conditions. Instead, these genes were significantly upregulated under erastin treatment (Fig. 3b), suggesting that under ferroptotic stress conditions, NRF2 is further activated when DJ-1 was knocked down. To further evaluate the role of NRF2 in DJ-1-regulated ferroptosis, we examined the effect of double KD of DJ-1 (#1 sequence of DJ-1 KD was used here) and NRF2 on the ferroptotic cell death induced by erastin (Fig. 3c). As shown in Fig. 3d, DJ-1 and NRF2 double KD further augmented the erastin-triggered cell death when compared to that of the single KD groups, excluding the possibility that NRF2 is involved in DJ-1-regulated ferroptosis.

**DJ-1 preserves transsulfuration-mediated GSH synthesis**. KD of DJ-1 promotes lipid ROS accumulation and cell death triggered by all four different classic ferroptosis inducers (erastin, sorafenib, RSL3, and ML210; Fig. 1, Supplementary Fig. 1). However, overexpression of DJ-1 in DJ-1$^{-/-}$ MEFs only reversed erastin and sorafenib-induced ferroptotic cell death, but not RSL3 and ML210-induced lipid ROS, and ferroptotic cell death (Fig. 2, Supplementary Fig. 2f, g). Since both RSL3 and ML210 are reported to bind to and inhibit GPX4 to trigger ferroptosis[7], our data suggested that DJ-1 might work on the upstream of GPX4.

Since SLC7A11 (also known as xCT) is the molecular target of erastin[1] and sorafenib[22], we next asked whether DJ-1 KD affects the expression and function of SLC7A11. As shown in Supplementary Fig. 3a, the SLC7A11 protein level was not significantly changed in DJ-1 KD H1299 cells under either basal or erastin treatment conditions. Moreover, there was no difference in SLC7A11 function between control and DJ-1 KD cells (Supplementary Fig. 3b). Additionally, we also found that DJ-1 KD didn't affect the intracellular cysteine levels (Supplementary Fig. 3c). Taken together, these results clearly rule out the possibility that DJ-1 regulates ferroptosis through SLC7A11.

In searching for other potential mechanisms, we noticed several interesting issues: (i) GSH is required for the proper function of GPX4 (ref. [23]); (ii) DJ-1 KD significantly reduced GSH levels compared to that of the control KD after erastin treatment (Fig. 4a). Moreover, a decrease in GSH was also observed in DJ-1 KO subclones with a better effect size due to strong DJ-1 inhibition (Supplementary Fig. 3d); and (iii) pretreatment of cells with exogenous GSH or N-acetylcysteine (NAC) resulted in complete reversal of erastin-induced lipid ROS accumulation (Fig. 4b) and cell death (Fig. 4c) in both control and DJ-1 KD cells. Therefore, we concluded that DJ-1 might affect the process of GSH synthesis.

Glutamate–cysteine ligase (γ-GCS) and glutathione synthetase (GSS) are the rate limiting enzymes for glutathione biosynthesis (Supplementary Fig. 3e)[24]. Although the expression of γ-GCS was slightly upregulated in response to erastin treatment, KD of DJ-1 did not block the upregulation of these two enzymes in H1299 cells (Supplementary Fig. 3f). Interestingly, when we treated cells with the transsulfuration pathway product L-homocysteine (Hcy), we observed that Hcy reversed the lipid ROS accumulation (Fig. 4d) and cell death (Fig. 4e) induced by erastin both in control and DJ-1 KD H1299 cells. In contrast, served as a system control, Hcy failed to reverse the lipid ROS accumulation induced by the cotreatment of erastin and the γ-GCS inhibitor buthionine sulphoximine (BSO), as expected (Supplementary Fig. 3e, g). Thus, DJ-1-silencing coordinates reduced GSH levels in ferroptosis, but its effect is not through regulation of the glutathione biosynthesis pathway.

Encouraged by the strong rescue effect of Hcy on erastin-induced ferroptosis in DJ-1 KD cells, we next focused on the transsulfuration pathway[25,26], which might be the predominant source of cysteine for GSH biosynthesis when xCT is inhibited. As shown in Fig. 4f, erastin treatment increased the mRNA levels of *CBS*, *MAT1*, and *CTH* (1.91-fold, 1.70-fold, and 2.35-fold, respectively), which are the known enzymes involved in the transsulfuration pathway (Fig. 4g). Moreover, we observed no significant difference in the mRNA level of these enzymes between control and DJ-1 KD cells after erastin treatment (Fig. 4f), suggesting that DJ-1 KD does not affect the expression level of these enzymes. Next, we treated cells with methionine (Met) and the transsulfuration pathway intermediates (S-adenosyl methionine (SAM) and S-adenosyl-L-homocysteine (SAH)) to perform a metabolite rescue assay. Unlike the effect of Hcy, neither Met nor the intermediates reversed the lipid ROS accumulation (Fig. 4d) or cell death (Fig. 4e) induced by erastin in DJ-1 KD cells. Thus, we hypothesized that the depletion of DJ-1 might disrupt the generation of Hcy from SAH in the transsulfuration pathway.

To test this, metabolic analysis was performed to semiquantitatively examine alterations of SAM, SAH, and Hcy levels in the transsulfuration pathway upon DJ-1 KD using mass spectrometry and carbon-13 labeling (Supplementary Fig. 4a). As shown in Supplementary Fig. 4b, c, we only observed a decrease in Hcy levels in DJ-1 KD cells (#1 sequence was used here, which had the best silencing effect of DJ-1 KD), as indicated by a reduction in Hcy $M + 1$ relative to control cells. To evaluate the levels of intracellular SAH and Hcy under ferroptotic conditions, we performed an enzyme-linked immunosorbent assay (ELISA) using DJ-1 KD H1299 and PANC1 cells with erastin treatment. As shown in Fig. 5a and Supplementary Fig. 4d, e, DJ-1 silencing significantly decreased Hcy levels, whereas the level of SAH, the upstream metabolite of Hcy, did not significantly change. Moreover, a decrease in Hcy was also observed in DJ-1 KO subclones with a better effect size (Fig. 5b). In line with this, a significant increase in Hcy levels and no difference in SAH level were observed in wild-type DJ-1, but not two DJ-1 mutants, reexpressed DJ-1$^{-/-}$ MEFs (Fig. 5c, Supplementary Fig. 4e, f).

To further confirm that the generation of Hcy from SAH is perturbed by DJ-1 KD, we treated cells with Met-null media for 24 h, followed by adding excess SAH into the cells, and specifically monitored the generation of Hcy from SAH (Fig. 5d). As expected, there was a significant increase (1.28-fold) in Hcy levels in control KD cells. However, DJ-1 KD cells showed no changes (1.01-fold) after adding SAH (Fig. 5e). Moreover, a significant increase in the Hcy levels was only observed in wild-type DJ-1 reexpressed DJ-1$^{-/-}$ MEFs after performing the same experiment (Fig. 5f, Supplementary Fig. 4g). Taken together, these findings suggest that DJ-1 KD promotes ferroptosis via downregulation of Hcy, which disturbs the generation of Hcy from SAH through the transsulfuration pathway.

**DJ-1 depletion impairs intracellular SAHH activity**. Since SAHH is the only known enzyme that catalyzes the hydrolysis of SAH to Hcy[27], we decided to examine the effect of DJ-1 on the enzymatic function of SAHH. HA-tagged SAHH was overexpressed in control, DJ-1 overexpressed, and DJ-1 KD HEK293T cells, and then the ectopic SAHH was purified from cells by immunoprecipitation (Fig. 5g). Interestingly, we found that the enzymatic activity of SAHH was dramatically increased by co-expressing DJ-1, but was decreased by DJ-1 KD (Fig. 5h). Our data also suggest that this effect relies on the proper functioning of DJ-1 (Supplementary Fig. 4h, i). To further validate this effect in a more physiologically relevant system, we examined the

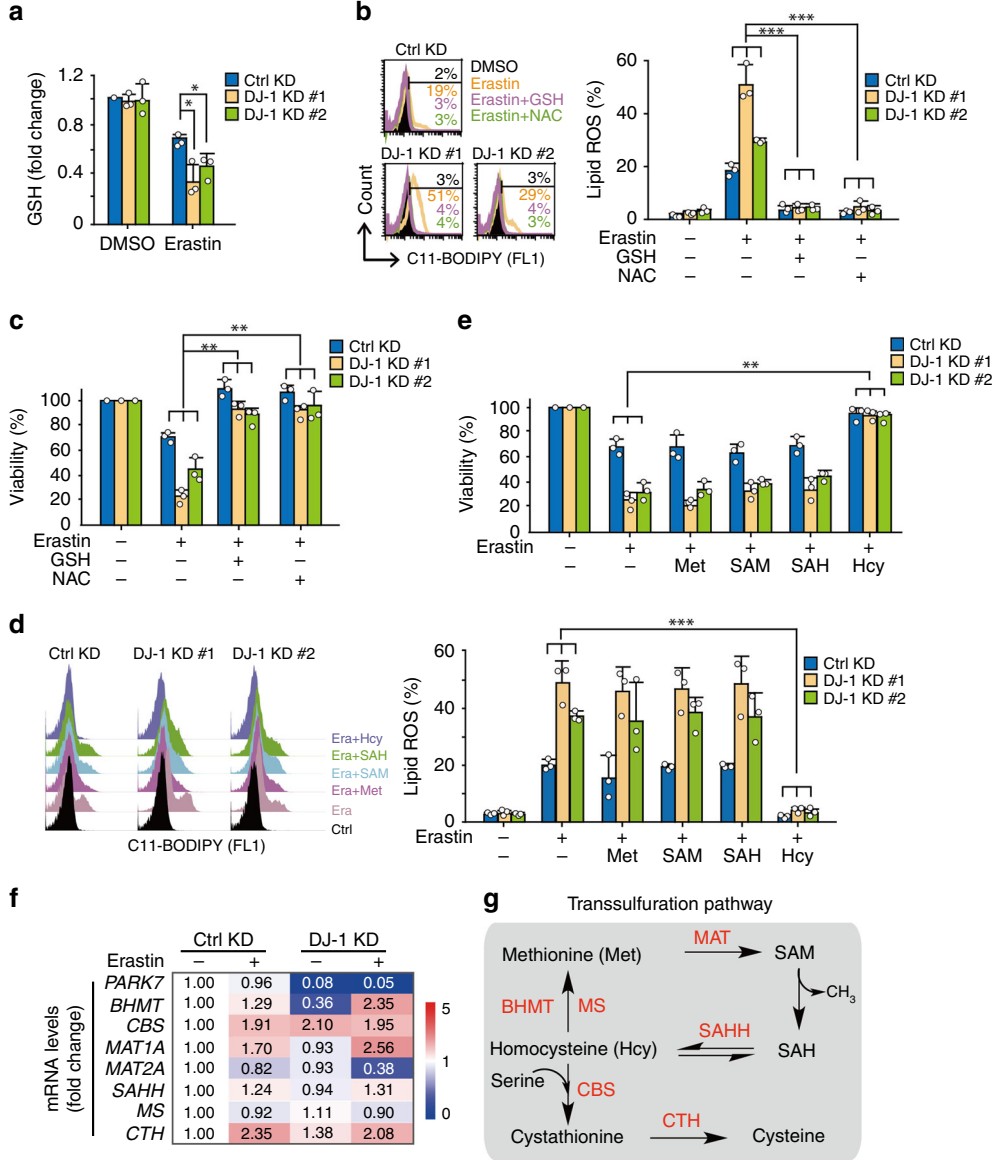

**Fig. 4 DJ-1 negatively regulates ferroptosis by impairing transsulfuration pathway-dictated GSH level. a** Indicated DJ-1 KD H1299 cells were treated with erastin (2 µM) for 6 h and intracellular GSH levels were examined. **b** Indicated DJ-1 KD H1299 cells were treated with erastin (2 µM) with or without GSH (0.5 mM) or NAC (0.5 mM) for 12 h, and lipid ROS production was assayed. **c** Cell viability was assayed in indicated DJ-1 KD H1299 cells treated for 36 h with erastin (2 µM) with or without GSH (0.5 mM) or NAC (0.5 mM). **d** Indicated DJ-1 KD H1299 cells were treated with erastin (2 µM) with or without indicated intermediates (Met, 0.5 mM; SAM, 0.5 mM; SAH, 0.5 mM; and Hcy, 0.5 mM) for 12 h, and lipid ROS production was assayed. **e** Cell viability was assayed in indicated DJ-1 KD H1299 cells treated for 36 h with erastin (2 µM) with or without indicated intermediates (Met, 0.5 mM; SAM, 0.5 mM; SAH, 0.5 mM; and Hcy, 0.5 mM). **f** Indicated DJ-1 KD H1299 cells were treated with erastin (2 µM) for 12 h, and the mRNA expression involved in transsulfuration pathway was assayed by qRT-PCR. The relative gene expression is normalized to β-actin and the number indicates the mean value from triplicates. **g** The schematic representation of transsulfuration pathway. Data shown represent mean ± SD from three independent experiments. Comparisons were made using the two-tailed, unpaired Student's t-test; *p < 0.05, **p < 0.01, ***p < 0.001.

effect of DJ-1 KD on the activity of endogenous SAHH (Supplementary Fig. 5a). DJ-1 KD significantly impaired the intracellular enzymatic activity of SAHH (Supplementary Fig. 5b). Thus, we showed that DJ-1 is required for maintaining the proper function of SAHH.

Given that suppression of SAHH expression by RNA interference also promoted erastin-induced accumulation of lipid ROS (Fig. 5i, j), which phenocopied the DJ-1 KD effect, it implies that DJ-1-regulated SAHH activity may be meaningful for preventing erastin-triggered ferroptosis. To test this hypothesis, we investigated whether overexpression of SAHH could reverse the increase in ferroptosis caused by DJ-1 KD. With simultaneous

overexpression of SAHH, H1299 cells were slightly, but significantly, insensitive to erastin treatment compared to that of DJ-1 KD cells (Fig. 5k–m). Notably, the reverse effect of SAHH overexpression was markedly enhanced when we additionally treated cells with SAH, the natural substrate of SAHH (Supplementary Fig. 5c, d). Thus, our data further suggest that SAHH participates in DJ-1-regulated ferroptosis.

**DJ-1 depletion enhances the SAHH/AHCYL1 interaction.** Next, we sought to understand how DJ-1 affects the activity of SAHH. Although our SAHH activity analysis documented that DJ-1

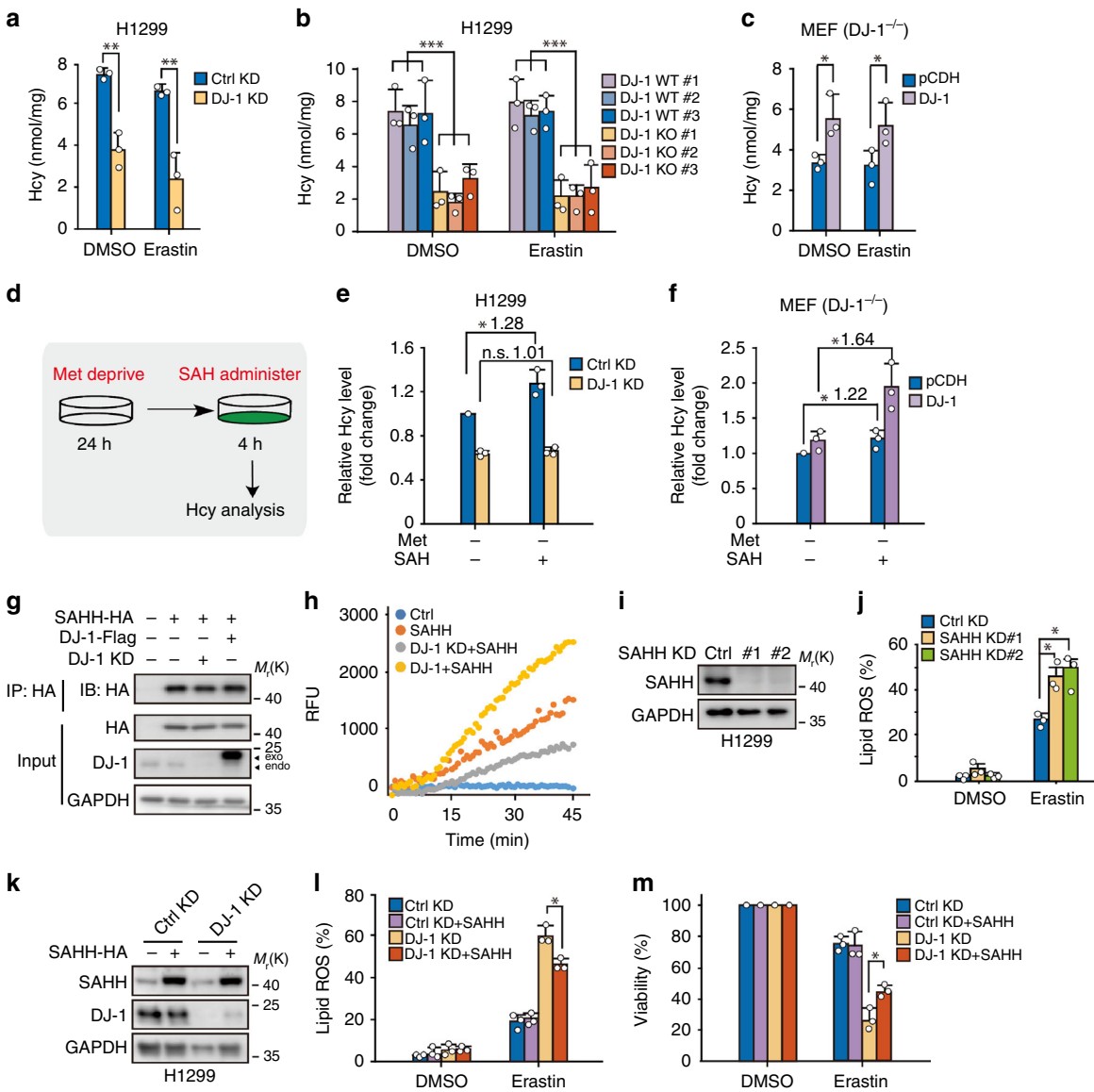

**Fig. 5 DJ-1 depletion disrupts the generation of Hcy from SAH via impairing intracellular SAHH activity. a–c** ELISA assays for the levels of endocellular Hcy. Indicated DJ-1 KD H1299 cells **a** and DJ-1 KO H1299 cells **b** were treated with erastin (2 μM) for 12 h, and the Hcy levels were assayed. Indicated MEFs **c** were treated with erastin (400 nM) for 12 h, and the Hcy levels were assayed. **d–f** Indicated cells were deprived from Met for 24 h, followed by adding the extra SAH to the cells for 4 h, and Hcy levels we detected by ELISA. **d** The schematic representation of experiment design. The relative Hcy levels in indicated H1299 cells **e** and MEFs **f** are shown. **g, h** DJ-1 increased the enzymatic activity of SAHH. **g** Indicated HEK293T cells with either DJ-1 overexpression or KD (#1 sequence of DJ-1 KD was used here) were further transfected with SAHH-HA plasmids. Cell lysates were immunoprecipitated with anti-HA antibody, followed by immunoblotting with anti-HA antibody. Independent experiments are repeated three times and representative data are shown. **h** The activity of ectopic SAHH from cells by immunoprecipitation was assayed as mentioned in Methods section. Independent experiments were repeated three times and representative data are shown. **i** Western blot analysis of SAHH expression in H1299 cells with SAHH KD. Independent experiments are repeated three times and representative data are shown. **j** Indicated SAHH KD H1299 cells were treated with erastin (2 μM) for 12 h, and lipid ROS production was assayed by flow cytometry using C11-BODIPY. **k** Western blot analysis of DJ-1 and SAHH expression in indicated H1299 cells (#1 sequence of DJ-1 KD was used here). Independent experiments are repeated three times and representative data are shown. **l** Indicated DJ-1 KD (#1 sequence) and SAHH overexpression H1299 cells were treated with erastin (2 μM) for 12 h, and lipid ROS production was assayed. **m** Cell viability was assayed in indicated cells treated with 2 μM erastin for 36 h. Data shown represent mean ± SD from three independent experiments. Comparisons were made using the two-tailed, unpaired Student's t-test; *p < 0.05, **p < 0.01, ***p < 0.001; n.s., no statistic difference.

determines intracellular SAHH activity (Fig. 5h), this effect could not be reconstituted by using recombinant DJ-1 and SAHH (Supplementary Fig. 5e). We thus reasoned that DJ-1 might affect the activity of SAHH indirectly. We then tested whether DJ-1 influences the expression and posttranslational modification of SAHH. As shown in Supplementary Fig. 5f–i, DJ-1 did not change the protein level of SAHH or its reported posttranslational

modifications, such as acetylation and phosphorylation level[28,29], indicating that DJ-1 might indirectly affect the activity of SAHH though another partner protein. To identify the potential regulator, we focused on the interaction protein with DJ-1 by utilizing the proximity-dependent biotin identification (BioID) method, in which the mutant form of biotin ligase (BirA*) releases the activated biotin and thus labels the proteins in proximal proteins with

biotin[30]. The DJ-1 and BirA* were fused and introduced into HEK293T cells. The cells were treated with biotin, and the biotin-labeled proteins were pulled down with streptavidin and subsequently analyzed by mass spectrometry (Fig. 6a). As shown in Fig. 6b and Supplementary Table 1, there were 123 proteins were specifically identified in the biotin-treated BirA*-DJ-1 cells. Among these proteins, SAHH-like protein (AHCYL) caught our

attention, not only because it ranked in the top ten based on the scores generated by analytic software, but also because it plays a dominant negative role in the regulation of SAHH activity[31,32].

To date, two AHCYLs, AHCYL1 and AHCYL2, have been identified in mammals, and these have largely overlapping functions with high similarity in gene structure (92% identical and 97% similar)[32]. The sequences of AHCYL protein sequences

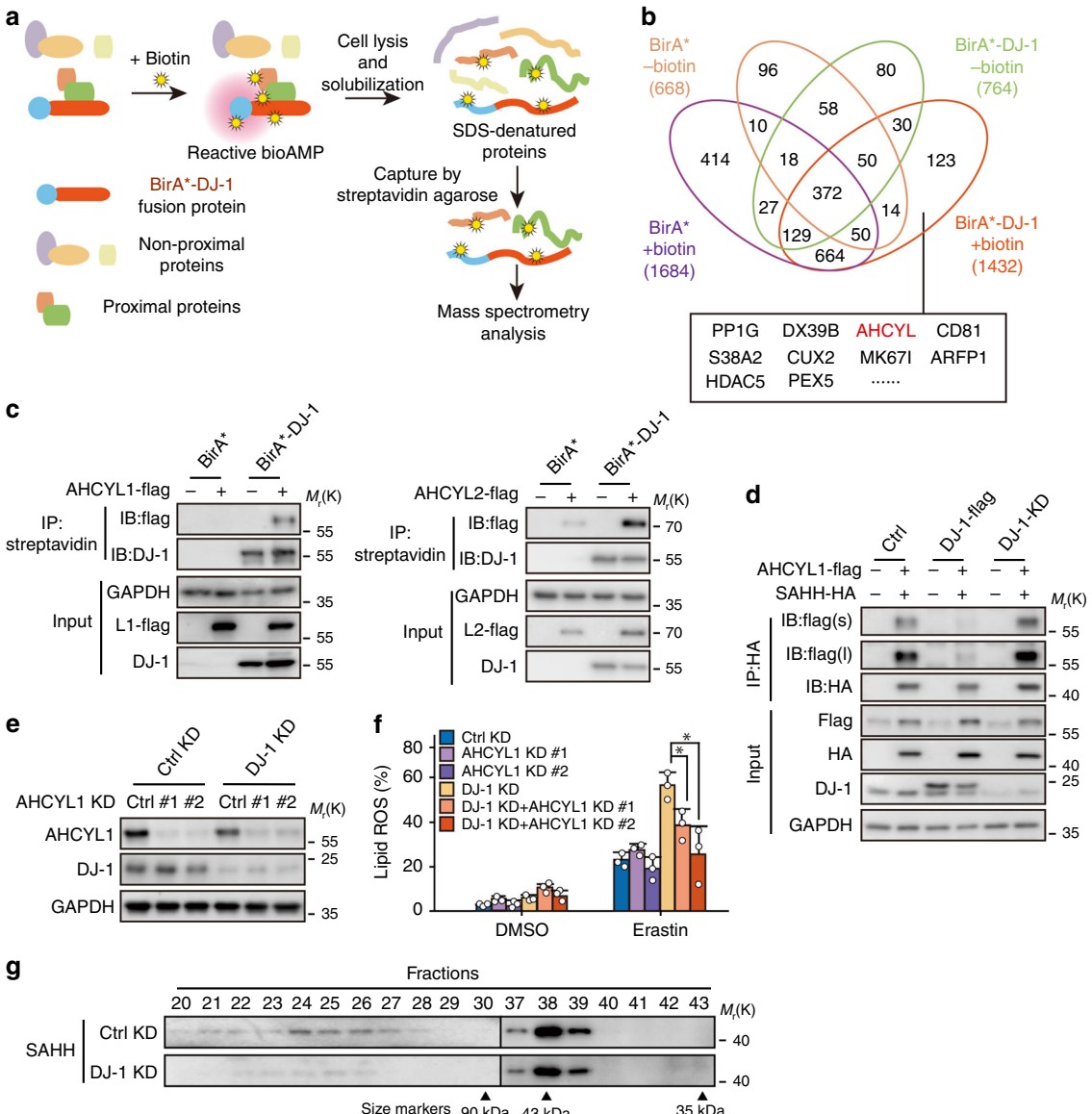

**Fig. 6 DJ-1 depletion impairs SAHH tetrameric structure via enhancing the SAHH/AHCYL1 interaction. a** The scheme of BioID assay for identifying the interaction proteins of DJ-1. **b** The results of BioID assay. The numbers indicate the proteins identified from each group by mass spectrometry, and the top ten proteins specifically identified in biotin-treated DJ-1-BirA* samples were listed. **c** The interaction between AHCYL1 (the left view) or AHCYL2 (the right view), and DJ-1 in BirA* system was analyzed. HEK293T cells were transfected with indicated plasmids and treated with DMSO or 50 μM biotin for 24 h before harvested. Input and samples of IP-streptavidin were separated on SDS–PAGE gels and subjected to immunoblotting with anti-Flag and anti-DJ-1 antibodies. Independent experiments are repeated three times and representative data are shown. **d** The interaction between AHCYL1 and SAHH under different DJ-1 level was analyzed. Indicated HEK293T cells stably expressing AHCYL1 and SAHH were harvested for immunoprecipitation and subjected to immunoblotting with anti-Flag and anti-HA antibodies (#1 sequence of DJ-1 KD was used here). 's' represents short exposure; 'l' represents long exposure. Independent experiments are repeated three times and representative data are shown. **e** Western blot analysis of DJ-1 and AHCYL1 expression in indicated H1299 cells (#1 sequence of DJ-1 KD was used here). Independent experiments are repeated three times and representative data are shown. **f** Indicated DJ-1 KD (#1 sequence) and AHCYL1 KD H1299 cells were treated with erastin (2 μM) for 12 h, and lipid ROS production was assayed. Data shown represent mean ± SD from three independent experiments. Comparisons were made using the two-tailed, unpaired Student's t-test; *p < 0.05. **g** Western blot analysis of the fractions from size-exclusion chromatography experiment, showing less tetrameric SAHH complex (fraction 23–28) in DJ-1 KD H1299 cells (#1 sequence of DJ-1 KD was used here). Independent experiments are repeated three times and representative data are shown.

identified from the BioID assay was "GIVEESVTGVHR", which is involved in both AHCYL1 and AHCYL2. We thus first tested whether AHCYL1 or AHCYL2 interacted with DJ-1 predominately. AHCYL1-Flag and AHCYL2-Flag were introduced into HEK293T cells, respectively, and analyzed by immunoprecipitation. As shown in Fig. 6c, both AHCYL1 and AHCYL2 were labeled with biotin as a result of interaction with BirA*-DJ-1; however, some background signal was detected in the AHCYL2 groups, suggesting that AHCYL1 was relatively specific.

It has been documented that SAHH functions as a tetramer with the cofactor NAD$^+$/NADH bound to each subunit, and the heteromultimerization of AHCYL and SAHH can decrease SAHH activity[31,32]. Thus, we hypothesized that DJ-1 changes the formation of heteromultimers of AHCYL (AHCYL1 or AHCYL2) and SAHH, and then impacts the tetramerization of SAHH. To test this hypothesis, we first determined the changes in the interaction between SAHH and AHCYL by manipulating the DJ-1 level in cells. As shown in Fig. 6d and Supplementary Fig. 6a, b, a high level of DJ-1 by DJ-1-flag transfection attenuated the interaction between AHCYL1 and SAHH, and DJ-1 silencing increased their heteromultimerization. However, the interaction between AHCYL2 and SAHH was not changed significantly with the manipulation of DJ-1 expression (Supplementary Fig. 6c). Importantly, the increase in lipid ROS production in the indicated DJ-1 KD H1299 cells was only inhibited after simultaneously AHCYL1 silencing (Fig. 6e, f, Supplementary Fig. 6d), but not after AHCYL2 silencing (Supplementary Fig. 6e–g). Additionally, as shown in Supplementary Fig. 6h, the interaction between AHCYL1 and SAHH is also DJ-1 functioning dependent, which is consistent with our cellular data mentioned above. Thus, our data confirm that the regulation of SAHH activity by DJ-1 was mediated by AHCYL1.

Last, by utilizing the well-established protein crosslinker disuccinimidyl suberate (DSS), we were able to detect both the tetrameric SAHH complex (size ~160 kDa) in addition to the monomer subunit (size ~43 kDa; Supplementary Fig. 6i). Of note, DJ-1 KD significantly impaired the formation of the SAHH tetramer (Supplementary Fig. 6i). In line with this, we observed that the tetrameric SAHH complex was dramatically impaired in DJ-1 KD cells by using size-exclusion chromatography (Fig. 6g). Moreover, DJ-1 overexpression also increased tetrameric SAHH complex formation in HEK293T cells (Supplementary Fig. 6j). Taken together, our data further suggest that DJ-1 depletion impairs SAHH tetrameric structure by enhancing the SAHH/AHCYL1 interaction, which further explains how DJ-1 determines SAHH activity.

**Suppression of DJ-1 enhances ferroptosis in vivo.** To determine whether suppression of DJ-1 enhances the antitumor activity of erastin in vivo, we evaluated the antitumor activity of erastin in a xenograft nude mouse model generated with H1299-shRNA-Control or H1299-shRNA-DJ-1 cells. Beginning at day 12, vehicle or 30 mg/kg piperazine erastin (PE) was administered to the mice through tail vein injection once every other day for another 16 days. As illustrated in Fig. 7a, b and Supplementary Fig. 7a, in the shRNA-Control groups, 30 mg/kg PE weakly decreased tumor weight, but it had no obvious effect on tumor volume or treatment/control (T/C) value (the ratio of relative tumor volume (RTV) in treated versus control mice) when compared with that of vehicle controls. Of note, DJ-1 depletion significantly enhanced the sensitivity of H1299 cells to PE, as the tumor volumes were significantly inhibited from days 7 to 16, the tumor weight decreased by 45.49%, and the T/C value was only 0.36. Next, we examined how DJ-1 depletion exhibits a synergistic effect with PE on tumor development and mainly focused on

three aspects: proliferation, apoptotic death induction, and ferroptosis. We isolated the tumor tissues, and subsequently performed the immunohistochemistry analysis for Ki67, which is a well-characterized marker expressed in proliferating cells. As shown in Fig. 7c, strong expression of Ki67 was detected in tumor tissues of all groups. Quantification analysis indicated that there was no significant difference of Ki67 expression among the groups (Supplementary Fig. 7b), implying that DJ-1 depletion did not impact tumor cell proliferation in vivo. A terminal deoxynucleotidyl transferase dUTP nick-end labeling (TUNEL) assay showed that few TUNEL-positive cells were detected in the four groups compared with that of the positive control group (Fig. 7d, Supplementary Fig. 7c), suggesting that apoptosis was not involved in the synergistic effect of DJ-1 depletion with PE. Indeed, quantitative reverse transcription polymerase chain reaction (qRT-PCR) analysis of the expression of prostaglandin-endoperoxide synthase-2 (PTGS2), a marker for the assessment of ferroptosis in vivo[7], indicated that KD of DJ-1 increases ferroptosis as PTGS2 levels increased (Fig. 7e). Moreover, the lipid ROS product 4-HNE was also significantly increased in the DJ-1 KD group after PE treatment as expected (Fig. 7f). Of note is the fact that Hcy levels were significantly lower in DJ-1 KD tumors compared with that of the control group under both basal and PE treatment conditions (Fig. 7g), which is in line with our cell-based results (Fig. 5). Thus, our data suggest that DJ-1 silencing enhances the sensitivity of H1299 cells to PE-induced tumor growth arrest, showing that DJ-1 plays a critical role in ferroptosis in vivo.

## Discussion

Accumulating studies have shown the molecular mechanisms and signaling pathways of ferroptosis, which lay the foundation for ferroptosis-based cancer therapy. These studies suggest that p53 inhibits cysteine uptake and thus sensitizes cells to ferroptosis, which forms a new point of view to explain the tumor-suppressive function of p53 (refs. [9,33–35]). NRF2 protects against ferroptosis by transcriptional activation of genes involved in lipid ROS and iron metabolism[8]. This leads to resistance of sorafenib in hepatocellular carcinoma cells. Additionally, studies have shown that HSF1-HSPB1 signaling negatively regulates erastin-induced ferroptosis in a variety of cancer cells by reducing iron-mediated production of lipid ROS (ref. [36]). The work presented here identifies DJ-1 as a negative regulator of ferroptosis in cancer cells. Suppression of DJ-1 expression, promotes ferroptotic cell death in several cancer cells, with exaggerated ferroptotic events induced by erastin, sorafenib, RSL3, and ML210. Given that DJ-1 protects tumor cells from various oxidative agents and determines the sensitivity of cancer cells to chemotherapy-induced apoptosis[37–42], our current study further expands the application of DJ-1 inhibition in ferroptosis-based cancer therapy.

In the case of ferroptosis, the transsulfuration pathway and its downstream glutathione biosynthesis pathway are expected to remedy the GSH level to remove the accumulation of lipid ROS. However, most efforts have focused on the critical role of the glutathione biosynthesis pathway in ferroptosis[7,8]. Little effort has been made to address the importance of the transsulfuration pathway under ferroptotic stress condition[43]. In the current study, we surprisingly found that the transsulfuration pathway, but not the glutathione biosynthesis pathway as previously reported, is tightly regulated by DJ-1 in negatively determining ferroptosis. This further emphasizes that manipulation of the transsulfuration pathway could also be considered for treating ferroptosis-associated diseases.

Of interest, the effect on SAHH by DJ-1 is a widespread fact. As our data show, even without erastin treatment, DJ-1 KD

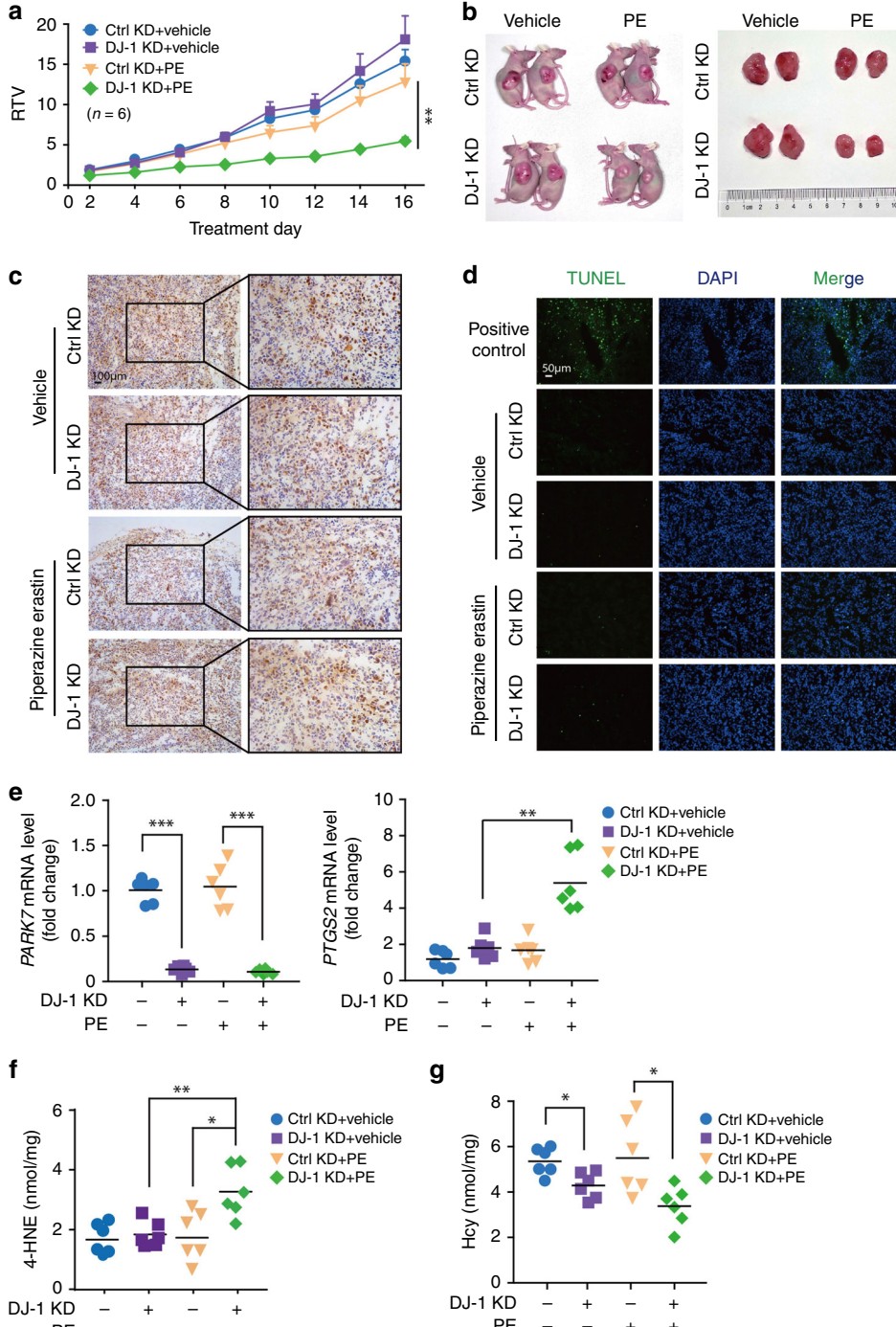

**Fig. 7 Suppression of DJ-1 enhances ferroptosis-based antitumor therapy in vivo. a** The mice were injected subcutaneously with indicated H1299 cells (#1 sequence of DJ-1 KD was used here) and treated with piperazine erastin (PE; 30 mg/kg, once every other day) through tail vein injection for 16 days. The tumor volume was recorded daily, and the relative tumor volume (RTV) was calculated ($n = 6$). RTV was shown as the mean ± SEM. **b** Representative images of tumors in each treatment group. **c** Representative immunohistochemical image of Ki67 in tumor sections are shown. The experiment was repeated twice independently with similar results. **d** The TUNEL-positive ratios of tumor regions were analyzed. And the representative images of TUNEL staining in the tumor tissues in different groups are shown. The experiment was repeated twice independently with similar results. **e** The mRNA expression of *PARK7* and *PTGS2* in indicated H1299 cells was assayed by qRT-PCR. The relative gene expression was normalized to β-actin. **f** The 4-HNE levels in indicated tumors were assayed by ELISA. **g** The Hcy levels in indicated tumors were assayed by ELISA. $n = 6$ biologically independent samples per group. Comparisons were made using the two-tailed, unpaired Student's *t*-test; *$p < 0.05$, **$p < 0.01$, ***$p < 0.001$.

distinctly decreased Hcy levels. However, this effect did not subsequently cause the depletion of GSH, until the cystine import was blocked. As we illustrated in Fig. 8, there are two routes to generate the intracellular cysteine in cells: one is the cystine import though SLC7A11 and the other one is the transsulfuration pathway. Under basal conditions, the cystine import plays the dominate role in the GSH precursor cysteine generation. Therefore, DJ-1 depletion can markedly inhibit GSH levels only when

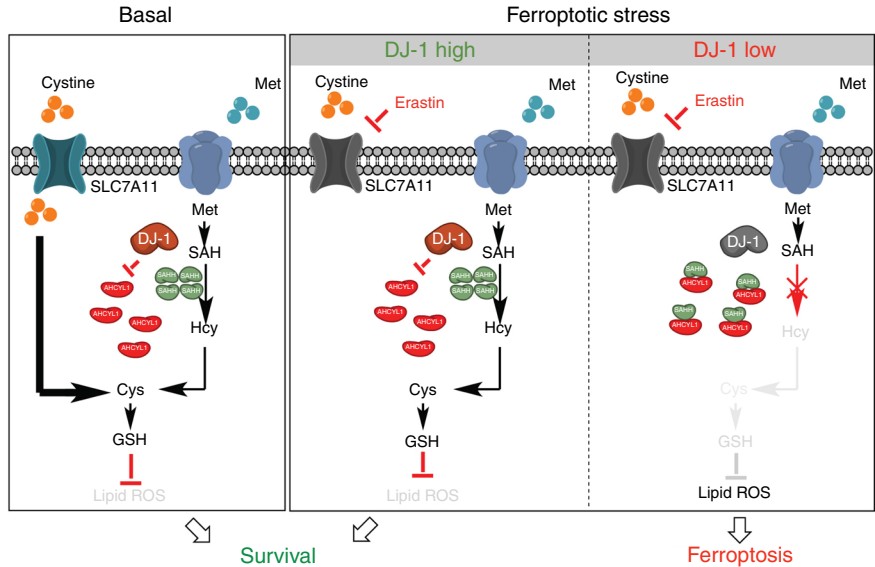

**Fig. 8 Scheme for the effect of DJ-1 on negatively regulating ferroptosis.** There are two routes to generate the intracellular cysteine in cells: one is the cystine import though SLC7A11 and the other one is the transsulfuration pathway. Under basal conditions, the cystine import plays the dominate role in the GSH precursor cysteine generation. When the cystine import is blocked by erastin, the DJ-1 preserved transsulfuration pathway still could provide essential cysteine synthesis (refer to DJ-1 high with ferroptotic stress). DJ-1 depletion can markedly inhibit GSH levels only when the cystine import is blocked (refer to DJ-1 low with ferroptotic stress). The DJ-1 KD distinctly decreases Hcy levels by impairing SAHH activity via enhancing its interaction with AHCYL1. Decreased level of Hcy subsequently causes the depletion of GSH and eventually enhances ferroptotic cell death.

the cystine import is blocked. This model also explained why reexpression of DJ-1 in DJ-1 KO MEFs only reversed the ferroptosis induced by erastin and sorafenib, but not GPX4 inhibitors. Since glutamate is the endogenous inhibitor of xCT (refs. [44,45]), and higher serum concentrations of glutamate have been observed in Parkinson's disease patients[46], our findings further speculate that DJ-1 mutant neuronal cells experience high levels of ferroptosis, which might establish a potential mechanism via which DJ-1 could regulate early-onset recessive Parkinson's disease.

Though it has been documented that DJ-1 positively dictates intracellular GSH levels by stabilizing NRF2 (ref. [11]) and mRNAs involved in glutathione metabolism[47], we didn't observe these changes in our experimental system. Since, we tested this phenotype in several cell lines, the possible reason for this contradictory finding can't be explained by cell line specificity. Since glutathione is still a general antioxidant reagent, and our mechanistic dissection revealed that DJ-1 maintains glutathione synthesis through preserving homocysteine production under cystine import deficient conditions, the manifestation of DJ-1's ferroptosis protective role could be highly context dependent. The results of our study revealed that SAHH contributes to the DJ-1-mediated lipid ROS response and ferroptosis, which provide direct evidence for a unreported antioxidant mechanism of DJ-1. Of interest, KD (or KO) of DJ-1 doesn't abolish intracellular SAHH enzymatic activity, which further implies that the effect of DJ-1 on SAHH is finely regulated.

It's worth to mention that the effect of DJ-1 KD/KO on ferroptosis under GPX4 inhibition condition might not be perfectly explained by DJ-1-SAHH axis. Because there is no change of GSH level without blocking cystine uptake, and it will not affect the effect of GPX4 inhibitors. Though the exact mechanism is still not understand, DJ-1 might also affect the effect of GPX4 inhibitors through other mechanisms. Especially, at most recently, two back-to-back papers demonstrated that ferroptosis suppressor protein 1 (FSP1) functioned as a glutathione-independent ferroptosis suppressor, suggesting there could be more suppressors and regulatory ways to coordinately dictate ferroptosis[48,49].

In our mechanistic study, we utilized the BioID assay to identify the potential regulator involved in DJ-1-dictated SAHH activity. And we identified that ACHYL1 is a interaction protein of DJ-1, which also has been known for negatively regulating SAHH activity via forming a heteromultimer complex. Interestingly, our data prove that DJ-1 determines the interaction between SAHH and ACHYL1, and subsequently affects the SAHH tetrameric structure. Although our current study hasn't clarified how DJ-1 dictates the interaction between SAHH and AHCYL1, it can't be explained by DJ-1 hijacking the AHCYL1, which in turn alters the heteromultimer complex formation. Because we only detected the interaction between AHCYL1 and SAHH, but not the interaction between AHCYL1 and DJ-1, through co-immunoprecipitation assay, suggesting the DJ-1/AHCYL1 interaction is much weaker than AHCYL1/SAHH heteromultimer complex. Given that DJ-1 is characterized as a class III glycase most recently[50,51], it's possible that DJ-1 enzymatically works on AHCYL1 and changes its property. Illustrating the mechanism behind will be a focus for future investigations. Another important point is that comparing to previous co-precipitation method for identifying the interaction proteins of DJ-1, the BioID assay allows to further capture transiently interacting proteins. This approach unsurprisingly detected hundreds of non-documented DJ-1-interacting proteins. It should be noted that based on our experience, many factors could cause the high false positive rates in interactome study (both co-precipitation method and BioID assay). Thus, without further biochemical validation, none of the hits identified should be considered as true functional DJ-1-interacting protein.

Taken together, we identified DJ-1 as a regulator of ferroptosis. As a defensive protein, DJ-1 maintains the function of the transsulfuration pathway, which is activated in response to ferroptosis. This regulation may be restrained in DJ-1 depletion, leading to a severe cell death triggered by ferroptotic stress (Fig. 8). These observations enhance our understanding of the mechanisms involved in ferroptosis and reveal that DJ-1 performs out its antioxidant function partially by regulating the transsulfuration pathway.

## Methods

**Antibodies and reagents**. The antibodies to DJ-1 (#5933), p-p38 (#9216), p38 (#8690), AHCYL1 (#94248), SLC7A11 (#12691), Acetylated-Lysine (#9441), and Phospho-(Ser/Thr) Phe (#9631) were obtained from Cell Signaling Technology. The primary antibody against NRF2 (sc13032), γ-GCS (sc390811), GSS (sc166882), SAHH (sc271389), and Ki67 (sc15402) were obtained from Santa Cruz Biotechnology. The primary antibody against GAPDH (db106), HA (db2603), and Flag (db7002) were obtained from Diagnostic Biosystems. GSH (G4251), NAC (A7250), DL-Met (M9500), L-glutamine (G3126), SAM (A4377), DL-homocysteine (H4628), SAH (A9384), and BSO (19176) were obtained from Sigma-Aldrich. Erastin (S7242), Fer-1 (S7243), and RSL3 (S8155) were obtained from Selleck Chemicals. Sorafenib (S125098) was obtained from Aladdin. ML210 (GC18705) was obtained from Glpbio. PE (HY100887) was obtained from Med-Chem Express. DMEM (GIBICO, #12800), RPMI-1640 (GIBICO, #31800), and DMEM without Met (#21013024) were obtained from Thermo Fisher Scientific.

**Cell culture**. The H1299, A549, PANC1, H292, H838, 786-O, KHOS, A2780, and HEK293T cell lines were purchased from the Shanghai Institute of Biochemistry and Cell Biology (Shanghai, China). The MEF DJ-1 KO cells were generated from day 13.5 embryos of DJ-1 KO mice (B6.Cg-*Park7*tm1Shn, #006577, The Jackson Laboratory) according to standard procedures[52]. H1299, H292, H838, A549, and 786-O cells were cultured in RPMI-1640 medium. PANC1, KHOS, A2780, HEK293T, and MEF DJ-1 KO cells were cultured in DMEM medium. All media were supplemented with 10% fetal bovine serum (FBS; Hyclone, SV30160.03, GE Healthcare), 100 units per mL penicillin and 100 μg/mL streptomycin. All cells were incubated at 37 °C in a humidified atmosphere of 5% CO$_2$. All cell lines used were authenticated by short tandem repeat (STR) profiling. The cell lines were monitored for mycoplasma contamination every 6 months.

**Lentivirus transduction**. The lentiviral shRNA vector pLKO.1-puro (SHC001) was obtained from Sigma-Aldrich and pCDH-EF1-Puro plasmid was obtained from System Biosciences. The pLKO.1-shDJ-1 #1/#2, pLKO.1-shNRF2, pLKO.1-shSAHH, pLKO.1-shAHCYL1, pLKO.1-shAHCYL2, pCDH-DJ-1-WT and Mut, pCDH-SAHH-HA, pCDH-AHCYL1-Flag, and pCDH-AHCYL2-Flag were constructed using the primers listed in Supplementary Table 2. Lentivirus was produced by transfecting HEK293T cells with pCMV-R8.91 (packaging vector), pMD2-VSVG (envelope vector), and shRNA plasmids or pCDH plasmids by Lipofectamine 2000 (#11668019, Invitrogen). Virus-containing medium was harvested 48 h after transfection. For infection, cells were grown in six-well plates at 30–40% confluency, and 0.1–1 mL of each virus was added with 1 μL polybrene (6 mg/mL).

**CRISPR/Cas9-mediated KO of DJ-1**. CRISPR gRNAs were designed by http://crispor.tefor.net/. The annealed gRNA1 targeting exon 2 and gRNA2 targeting exon 3 were inserted into Bbs1-digested pSpCas9(BB)−2A-GFP (PX458) plasmid (Addgene plasmid #48138) to generate PX458-PARK7 gRNA1 and PX458-PARK7 gRNA2. The plasmids were sequenced to verify the sequence. The gRNA oligonucleotide sequences are as follows:

PARK7 gRNA1:
Sense (5′ to 3′): CACCGTAGATGTCATGAGGCGAGCT
Antisense (5′ to 3′): AAACAGCTCGCCTCATGACATCTAC
PARK7 gRNA2:
Sense (5′ to 3′): CACCGAGTACAGTGTAGCCGTGATG
Antisense (5′ to 3′′): AAACCATCACGGCTACACTGTACTC

To generate DJ-1 KO H1299 cells, H1299 cells were electroporated with PX458-PARK7 gRNA plasmids in electrotransfection buffer (120 mM KCl, 0.15 mM CaCl$_2$, 10 mM K$_2$HPO$_4$, 25 mM HEPES, 2 mM EGTA, 5 mM MgCl$_2$, 2 mM ATP, 5 mM glutathione, pH = 7.6) on a nucleic acid transfection apparatus (Amaxa Nucleofector II, Amaxa) with X-001 program. Clones derived from single DJ-1 KO cells were obtained by fluorescence-activated cell sorting (FACSAria II, BD) in a 96-well plate and the successfully edited clones were verified using western blot. The cDNA was extracted to perform PCR and was sequenced to verify the sequence. The primers are as follows:
Forward: 5′-ATGGCTTCCAAAAGAGCTCT-3′
Reverse: 5′-CTAGTCTTTAAGAACAAGTG-3′

**Cell viability assay**. Cell viability was typically assessed in 96-well format by Cell Counting Kit-8 (CCK-8; HY-K0301, MedChem Express). When added to cells, WST-8 [2-(2-methoxy-4-nitrophenyl)-3-(4-nitrophenyl)-5-(2, 4- disulfophenyl)] is modified by the reducing environment of viable cells and turns orange (WST-8 formazan) in color. Briefly, simply added 10 μL of the CCK-8 solution to each well of the plate, incubated for 1–4 h, and the absorbance at 450 nm was measured on a microplate reader (Multiskan spectrum 1500, Thermo). Cell viability under test conditions was reported as a percentage relative to the negative control.

**Analysis of lipid ROS production**. The day before the experiment, 150,000 cells/well were seeded in six-well dishes. The day of the experiment, cells were treated with test compounds for the indicated times, harvested by trypsinization, resuspended in 500 μL phosphate-buffered saline (PBS) containing 2 μM C11-BODIPY (581/591) (#D3861, Invitrogen), and incubated for 30 min at 37 °C in a tissue culture incubator. Cells were then resuspended in 500 μL of fresh PBS, strained through a 40 μm cell strainer, and analyzed using a flow cytometer (FACSuite, BD Biosciences) equipped with a 488 nm laser for excitation. Data were collected from the FL1 channel (527 nm). A minimum of 10,000 cells were analyzed per condition. The data analysis was performed by using FlowJo Version 7.6 software.

**Transmission electron microscopy**. H1299-shRNA-Con or H1299-shRNA-DJ-1 cells were plated at 600,000 cells/dish in 100 mm tissue culture dishes. After 24 h, cells were treated with vehicle (dimethyl sulfoxide, DMSO) or erastin (2 μM) for 12 h. Cells were fixed with 2.5% glutaraldehyde in 0.1 mM phosphate buffer, followed by 1% OsO$_4$ for 2 h. After dehydration, the cells were embedded in epoxy resin. The ultrathin sections were made by an ultramicrotome and then stained with lead citrate and uranyl acetate, the distribution of mitochondria was observed using a transmission electron microscope (Philips Electronic Instruments, Mahwah, NJ, USA).

**GSH assay**. The relative GSH concentration in cell lysates was assessed using a kit from Nanjing Jiancheng (#A006-2) according to the manufacturer's instructions. The measurement of GSH used a kinetic assay, in which catalytic amounts (nmoles) of GSH caused a continuous reduction of 5, 5′-dithiobis (2-nitrobenzoic acid) to 5-thio-2-nitrobenzoic acid and the GSSG formed was recycled by glutathione reductase and NADPH. The yellow product (5-thio-2-nitrobenzoic acid) was measured spectrophotometrically at 420 nm. The values were normalized by protein concentration and presented as percentages relative to the negative control.

**Cysteine assay**. The intracellular cysteine concentration was assessed using a kit from Nanjing Jiancheng (#A126-1) according to the manufacturer's instructions, in which phosphotungstic acid was reduced to tungsten blue by cysteine, and the blue product was measured spectrophotometrically at 600 nm. The values were normalized by protein concentration.

**Cystine uptake assay**. The uptake of cystine was assayed by the release of glutamate from H1299 cells into the extracellular medium, detected using an Amplex Red glutamate release assay kit (#1883805). Cells were washed twice in PBS and incubated for an hour in Na$^+$-containing, glutamine-free media. A volume of 50 μL of medium per well was removed and transferred to a 96-well assay plate and incubated with 50 μL of a reaction mixture containing glutamate oxidase, L-alanine, glutamate-pyrurate transaminase, horseradish peroxidase, and Amplex Red reagents as per the manufacturer's protocol. The values were normalized to total cell number and presented as percentages relative to the negative control.

**Hcy and SAH assay**. Cells were lysed using a freeze thaw method (five cycles) in assay buffer, samples were frozen in liquid nitrogen and subsequently thawed at 37 °C. The lysate was then centrifuged at 14,000 × $g$ for 20 min at 4 °C to remove insoluble materials. The Hcy and SAH assay was carried out using an ELISA kit following the manufacturer's suggested protocol (Jiancheng, Nanjing, China). The ELISA data analysis was collected using Molecular Devices SpectraMax M5 plate reader software SoftMax Pro 6.3. The values were normalized by protein concentration.

**Metabolite analysis**. H1299 control, DJ-1 KD cells were seeded overnight in normal RPMI-1640 medium containing 10% FBS. The next day, medium was removed and cells were washed twice in PBS and then incubated for 24 h in medium lacking Met (#A1451701, Fisher Scientific). The next day, conditional medium was removed and cells were washed twice in PBS and then incubated for 4 h in EBSS (#24010043, Fisher Scientific) supplemented with 63 mg/L-Met-1-$^{13}$C (#490083,Sigma-Aldrich). Cells were then harvested, washed with PBS and flash frozen in liquid nitrogen. Cells were resuspended in 1 mL 80% methanol that was precooled to −80 °C, and subsequently freezed in liquid nitrogen and then thawed at room temperature. The lysate was transferred to a new tube after centrifuging at 1400 × $g$ for 20 min. The supernatant was dried by vacuum centrifugation, and then the dried samples were stored in −80 °C freezer. A total of $1.0 × 10^7$ cells in each group were used for sample analysis.

Samples were processed by Metabolomics Core Facility (Beijing). Briefly, samples were reconstituted in 200 μL 80% methanol and analyzed by liquid chromatography and mass spectrometry (LC–MS/MS) using a BEH amide column (1.7 μm, 2.1 × 100 mm, Waters) coupled to a TSQ Quantiva Ultra triple-quadrupole mass spectrometer (Thermo Fisher, CA), equipped with a heated electrospray ionization probe in positive ion mode. The elution was carried out at a flow rate of 0.25 mL/min over a multistage gradient (buffer A was 50% water with 10 mM ammonium formate and 50% acetonitrile, subsequently, pH = 3.0; buffer B was 95% acetonitrile and 5% water with 10 mM ammonium formate, pH = 3.0): 0–1.5 min, 2% A; 1.5–14 min, 98% A; 14–16.5 min, 98% A; 16.5–17 min, 2% A; and 17–20 min, 2% A. Column chamber and sample tray were held

at 35 °C and 10 °C, respectively. Data acquired in selected reaction monitoring mode in positive ion mode. Both of precursor and fragment ion were collected with resolution of 0.7 full-width at half-maximum, respectively. The source parameters are as follows: spray voltage: 3500 V; ion transfer tube temperature: 350 °C; vaporizer temperature: 300 °C; sheath gas flow rate: 30 Arb; auxiliary gas flow rate: 10 Arb.; and CID gas: 1.5 mTorr. Data analysis and quantitation were performed by the software Xcalibur 3.0.63 (Thermo Fisher, CA). The retention time of SAM, SAH, and Hcy were determined to be 8.63 min, 7.9 min, and 1.2 min, respectively based on standard samples. The values shown in the results were percentages relative to the negative control shRNA. The experiment was done in biological triplicate.

**Proximity-dependent BioID assay.** We adopted a BioID technique by fusing DJ-1 with the mutant form of *Escherichia coli* biotin protein ligase (BiaA*)[53]. HEK293T cells were infected with lentivirus encoding DJ-1-BirA* and Ctrl-BirA*. On the day of experiments, 5 of 10-cm dishes of cells were treated with or without 50 μM biotin for 24 h. Cells were harvested, denatured by 4% sodium dodecyl sulfate (SDS) buffer (150 mM NaCl, 50 mM trimethylamine), and the biotin-labeled proteins were pulled down with streptavidin, separated on SDS–polyacrylamide gel electrophoresis (PAGE) gels and subsequently analyzed by mass spectrometry. Mass spectrometry data were acquired in Xcalibur 2.2 operation software (Thermo Fisher Scientific).

**Western blot analysis.** Cells were lysed with 1% NP40 buffer (50 mM Tris-HCl, 150 mM NaCl, 1% NP40, pH = 7.4, 0.1 mM sodium vanadate, 5 μg/mL leupeptin, and 0.1 mM phenyl methane sulfonyl fluoride) and incubated at 4 °C for 30 min. The lysate was then centrifuged at 14,000 × $g$ for 20 min at 4 °C to remove insoluble materials. Protein concentrations of whole-cell lysates were determined using the DC Protein Assay Kit (Bio-Rad, Hercules, CA, USA). The proteins were then electrophoresed in SDS–PAGE gels and transferred to polyvinylidene difluoride membranes (Millipore, Bedford, MA, USA). After blocking with 5% non-fat milk, the membrane was incubated with various primary antibodies overnight at 4 °C, and proteins were visualized by enhanced chemiluminescence detection (NEL103E001EA, PerkinElmer) by film exposure or AI600 (GE Healthcare) after incubation with the appropriate horseradish peroxidase-conjugated secondary antibodies.

**Immunoprecipitation.** Endogenous immunoprecipitation was performed by SureBeads Protein G Magnetic beads (#1614023, Bio-Rad). Briefly, beads were washed with T-PBS twice and then 2 μg of the primary antibodies or normal IgG was added at room temperature for 30 min with end-over-end mixing. After coupling, the complex washed with 1 mL PBS three times and the supernatants were discarded, subsequently 500 μL of cell lysates was added for 4 h at 4 °C with end-over-end mixing. Finally, the complex was washed with T-PBS five times and suspended in loading buffer and followed by western blot analysis.

Exogenous immunoprecipitation was performed by Anti-HA magnetic beads (B26201, Bimake). Briefly, beads were washed with T-PBS twice and then 500 μL cell lysate was added for 4 h at 4 °C with end-over-end mixing. Finally, the complex was washed with T-PBS five times and suspended in loading buffer and followed by western blot analysis.

**Quantitative real-time PCR analysis.** Total RNA isolation was carried out with RNAiso plus kit (#9109, TaKaRa) according to the manufacturer's instructions. And then first-strand complementary DNA synthesis was obtained using the Reverse Transcription System Kit (#AT311-03, TransGen Biotech) according to the manufacturer's instructions. The quantitative real-time RT-PCR analysis was performed by iTaq™ Universal SYBR Green Supremix (#172-5124, BIO-RAD). The reaction mixtures containing SYBR Green were composed following the manufacturer's protocol and then CT values were obtained using a qPCR platform (QuantStudio 6 Flex Real-Time PCR System, Thermo Fisher Scientific). Real-time PCR data was monitored using QuantStudio 6 Design and Analysis Software Version 2.3. The sequences of the primers used for the qPCR were listed in Supplementary Table 3.

**DSS-mediated cross-linking assays.** Cells were detached and incubated at room temperature (25 °C) for 30 min in 2 mM DSS crosslinker (suberic acid bis (N-hydroxysuccinimide ester), #S1885, Sigma), the reaction was quenched with 1 M Tris-HCl buffer, pH = 7.4, for 15 min at room temperature. Samples were analyzed by western blot.

**Size-exclusion chromatography.** A total of 500 μL of cell lysate (at ~2 mg/mL) were loaded onto a size-exclusion chromatography column (HiLoad 16/60 Superdex 200, GE Healthcare) in 20 mM Tris, pH = 8.0, 150 mM NaCl, at 0.3 mL/min with an ÄKTA Purifier (GE Healthcare). A volume of 100 mL eluent was collected at detailed density.

**SAHH enzymatic activity assay.** Endogenous SAHH and exogenous SAHH-HA were obtained by immunoprecipitation and suspended in SAHH assay buffer. The SAHH activity assays were performed using the Adenosylhomocysteinase Activity Fluorometric Assay Kit (K807-100, Biovision) according to the manufacturer's instructions. This kit allows quantitative detection of adenosine concentration as the readout for rSAHH's ability to hydrolyze SAH in vitro. Fluorescent signal of SAHH activity assay was collected using Molecular Devices SpectraMax M5 plate reader software SoftMax Pro 6.3.

**In vivo xenograft mouse study.** Tumors were established by a subcutaneous injection of shRNA (control or DJ-1 KD) transfected H1299 cells (1,000,000/200 μL) into BALB/c female athymic nude mice (5 weeks, National Rodent Laboratory Animal Resource, Shanghai, China). The mice were housed in individually ventilated cages and supplied with sterilized food, water, and bedding. The mice were maintained at environmental temperature and humidity ranges from 21 to 26 °C, and from 50% to 70%, respectively. Twelve days after injection, mice were randomly allocated into different groups and treated with vehicle (0.625% DMSO/99.375% HBSS (pH = 2)) or 30 mg/kg PE (tail intravenous injection, once every other day) for 16 days before the final tumor size was measured in all groups. Tumors were measured twice weekly, and volumes were calculated using the formula (length × width[2])/2. At the end of the experiment, the tumors were weighed, and the inhibition rate and T/C value (RTV of treatment/RTV of control) were calculated. Maintenance and experimental procedures for the mice studies were approved by Zhejiang University's IACUC (P-IACUC-004).

**Immunohistochemistry.** The tumor tissues were frozen in the freezing compound with isopentan and dry ice, and 8-μm-thick frozen sections were sliced with a cryomicrotome. Next, immunohistochemistry was performed using an automated immunostainer with a DAB detection kit according to the manufacturer's instructions, using anti-ki67 (dilution 1:50, sc-15402, Santa Cruz Biotechnology). Finally, the slides were counterstained with Mayer's hematoxylin for 2 min, washed with deionized water, drained, differentiated in 1% acid alcohol, dehydrated through graded alcohols, and cleared in xylene.

**TUNEL assay.** The TUNEL assay was used to evaluate the apoptotic response in frozen sections of tumor tissues using the one-step TUNEL Apoptosis Assay kit (C1088, Beyotime, China) according to the manufacturer's instructions. The positive control was used here for quality control, and 4′,6-diamidino-2-phenylindole was used to stain the nuclei. TUNEL-positive cells were counted in images. Imaging data was collected using a Leica microscope software Leica Application Suite Version 4.3.0.

**4-HNE assay.** For a ~30 mg piece of tissue, add ~300 μL complete extraction buffer (100 mM Tris, pH = 7.4, 150 mM NaCl, 1 mM EGTA, 1 mM EDTA, 1% Triton X-100, 0.5% sodium deoxycholate, supplement the cell extraction with phosphatase and protease inhibitor cocktails as described by manufacturer, and phenylmethylsulfonyl fluoride to 1 mM, immediately before use) to the tube and homogenize with an electric homogenizer (MagNA Lyser, Roche). The extract was then centrifuged for 30 min at 13,000 × $g$ at 4 °C to get supernatant. The 4-HNE assay was carried out using an ELISA kit following the manufacturer's suggested protocol (#ab238538, Abcam). The ELISA data analysis was collected using Molecular Devices SpectraMax M5 plate reader software SoftMax Pro 6.3. The values were normalized by protein concentration.

**Statistical analysis.** All statistical analyses were performed using Prism 5.0c (GraphPad Software). The number of biological replicates for each experiment is indicated in the figure legends. Sample sizes were chosen based on standard experimental requirements in molecular biology. Differences between means were determined using the two-tailed, unpaired Student's $t$-tests, and were considered significant at $p < 0.05$.

**Reporting summary.** Further information on research design is available in the Nature Research Reporting Summary linked to this article.

## Data availability

Raw data for all western blot and original data of all charts are provided as a source data file. The potential interaction proteins of DJ-1 based on BioID assay can be found in Supplementary Table 1. All the plasmids used in this manuscript, including the ones in BioID assay, will be available from the corresponding author upon reasonable request.

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

## Acknowledgements

This work was supported by grants from the National Natural Science Foundation of China (no. 81773757 to M.Y.; no. 81625024 to B.Y.; and no. 81402951 to J.C.), National Science & Technology Major Project "Key New Drug Creation and Manufacturing Program", China (2018ZX09711002-003-008 to H.Z.), Zhejiang Provincial Natural Science Foundation (no. Y18H310005 to J.C.). We thank Dr. Nicole Spiegelman for language editing.

## Author contributions

J.C., X.C., M.Yi., and B.Y. designed the research; J.C., X.C., M.Yi., Q.H., and B.Y. wrote the manuscript; J.C., X.C., L.J., B.L., and D.Z. performed the biochemical and cellular studies; J.C., X.C., L.J., and M.Yu. conducted the animal studies; J.C., X.C., H.Z., and B.Y. analyzed the results; and J.C., M.Yi., Q.H., and B.Y. directed the study.

## Competing interests

The authors declare no competing interests.
