## [Peer Review File · Nature Communications]

Reviewers' comments:

Reviewer #1 (Remarks to the Author): expertise in ferroptosis

In the present manuscript, Cao, Chen et al., describe that DJ-1 (PARK7), a gene associated with anti-oxidative response and Parkinson's disease, protects cells from lipid peroxidation-induced ferroptosis by promoting homocysteine (and subsequently glutathione) synthesis.

This work is of significance because of the close connection between DJ-1 and cancer and Parkinson's disease. A thorough understanding on the biochemical activity of DJ-1 is in need for the field to grasp how various cells employ DJ-1 to cope with oxidative stress. Additionally, this work has (indirect) implications on the pathogenesis of Parkinson's disease, in which context the involvement of ferroptosis is implicated but awaits to be clarified. This work is also timely as the glutathione synthesis pathway utilizing exogenous cystine (via SLC7A11) has been extensively characterized in the context of ferroptosis, whereas the contribution of the endogenous cysteine synthesis pathway (via methionine and the transsulfuration pathway) is less understood.

The presented work is novel in several ways. Besides establishing a link between DJ-1 and lipid peroxidation and ferroptosis, Cao, Chen et al., have 1) thoroughly examined the activities of multiple known DJ-1 variants, which to the referee's knowledge, have not been systematically characterized before; 2) thoughtfully dissected the involvement of DJ-1 in regulating the SAH-Homocysteine-Cys metabolic pathway, and concluded that DJ-1's ferroptosis protective activity lies in the SAH to homocysteine transition; 3) biochemically mapped the interactome of DJ-1 using unbiased mass spectrometry approach, and identified adenosylhomocysteinase-like 1 (AHCYL1) as the intermediate factor that negatively regulates SAAH activity. This set of mechanistic characterizations improve our understanding on how DJ-1 and the SAAH/AHCYL1 complex are connected to cellular redox homeostasis.

Additionally, the logic of the manuscript is generally sound and well-presented, the experiments are mostly well-controlled, major conclusions are assessed with both in vitro and in vivo models, and the final scheme is largely justified by the experimental results. Therefore, in this referee's opinion, this work is a strong candidate that matches the breadth and depth of publications in Nature Communications. Having said that, this study has the following major, minor and grammatical weaknesses that need to be addressed:

Major:

1. One major drawback of this work is that the extent of effects in many of the experiments appears to be modest or rather subtle. This comment applies to the results in viability analyses in Figures 1F, 2D, 2F, 4A,4C,4E, as well as in the later metabolic characterizations. It remains unclear, at this point, that whether the lack of stronger metabolic and ferroptotic effects following DJ-1 knockdown is due to: 1) the weak knockdown efficacy of the two shRNAs that were used as the only means of DJ-1 suppression throughout the study; 2) the generally weak ferroptotic responses and incomplete death of the cell lines examined, which may have other ferroptosis-suppressive mechanisms, e.g. low polyunsaturated lipid levels, low intracellular iron levels, that mask the effects of DJ-1; 3) the limited efficacy and specificity of Erastin in eliciting ferroptosis; 4) DJ-1's rather non-specific activity in diminishing cytosolic ROS; or 5) activation of compensatory pathways in response to DJ-1 suppression.

To substantiate that DJ-1 does play a critical role in ferroptosis protection, the following alternative approaches are recommended:

- a. Include an alternative loss of function approach to inhibit DJ-1 deeply, such as CRISPR/Cas9-based knockout, and assess how ferroptosis and homocysteine metabolism is affected in those contexts. This applies at least to Figures 1B, 1F, 4D and 4E.
- b. Present viability measurements of compound treated cells as wider range dose-response curves rather than selected concentrations, this applies at least to Figures 1F, 4D and 4E.

c. Use at least one alternative cell line (besides DJ-1^{-/-} MEF) that exhibits higher intrinsic sensitivity to ferroptosis, such as diffused large B-cell lymphoma (Yang, et al., Cell 2014), renal cell carcinoma (Miess et al., Oncogene, 2018; Zou et al., Nature Communications 2019), hepatocellular carcinoma (Sun et al., Hepatology 2016) or any other cell line of this kind, to test whether exogenous DJ-1 expression protects the cells from ferroptosis, and whether such cell death can be rescued by ferrostatin or liproxstatin.

2. While the authors started by speculating that DJ-1 may have a regulatory role in lipid ROS production and ferroptosis, the mechanistic dissection revealed that DJ-1 regulates glutathione synthesis through upregulating homocysteine production. Since glutathione is still a general antioxidant reagent that exhibits functions in and beyond ferroptosis, the manifestation of DJ-1's ferroptosis protective role could be highly context-dependent, i.e. stress-dependent. It is recommended that the authors go back to this point in the discussion, to clarify that DJ-1's anti-oxidative stress role is not specific for antagonizing lipid hydroperoxides, but rather general for both cytosolic and lipid ROS. Along this line, the assessment of general ROS levels, e.g. by H2-DCFDA in wildtype vs DJ-1 knockdown/knockout cells is recommended as a control.

3. (Optional) The in vivo experiment demonstrated a tumor suppressive effect of DJ-1 knockdown and piperazine erastin (PE) treatment, however, the connection of this effect to ferroptosis in vivo is only minimally justified by the usage of erastin, and the up-regulation of PTGS2, which can be induced by many cellular stress responses including inflammation, in addition to be upregulated in ferroptotic tissues. Hence a treatment arm with a lipophilic anti-oxidant that effectively and specifically blocks ferroptosis, such as liproxstatin (Friedmann Angeli et al., Nature Cell Biology 2014) is recommended for such in vivo experiments.

Minor:

1. The authors showed in Figures 2 and S1D that RSL3-induced ferroptosis in DJ-1^{-/-} MEF cells was not reversed by DJ-1 overexpression, and suggested that DJ-1 does not directly affect GPX4. It seems unclear to this referee that if DJ-1 inhibition does have a substantial effect on glutathione abundance, how would the inhibition of GPX4, which utilizes glutathione as a substrate, and is the only lipid hydroperoxide-specific glutathione peroxidase in the cell, is completely unaffected by DJ-1 inhibition. It's important and relevant to clarify this point, perhaps by using a more specific GPX4 inhibitor, ML210 (Eaton et al., BioRxiv (doi: <https://doi.org/10.1101/376764>)), and by presenting the viability results as wide range dose-response curves.

Grammatical:

Grammatical issues include but are not limited to the ones listed in the following, and assistance from professional text editing should be considered.

Line 116, "mutant" should be "mutants"

Line 117, "pathogenic associated" better to use "disease-associated", since not every mutation has a causal role in (Parkinson's) disease

Line 124 and thereafter, NRF2 and KEAP1 should be capitalized if referring to human proteins

Line 167, "was" needs to be deleted

Line 194, "to further confirm the generation" needs to be "to further confirm that the generation..."

Line 225, "SAHH is participated in..." should be "SAHH participates in ..."

Line 228, "BioID" may have been mistyped as "BiaID"

Line 254, the author may mean "relatively more specific" by "relatively specific"

Line 262, "slience" needs to be "silencing"

Line 288, "interested to ask" may better be "interested in asking"

Line 332, typo in "itu"

Line 353, "hijacks the" should be "hijacking"

Line 358, "in" should be deleted

Line 359, "Illustration" should be "Illustrating"

Line 361, "the transit binding proteins", "transiently interacting/bound proteins"

Line 362, "That's not surprisingly found hundreds of new potential DJ-1 interaction proteins." May better be "This approach unsurprisingly detected hundreds of new DJ-1 interacting proteins"
Line 366, "interaction" should be "interacting"
Figure 2D. Overlapped texts in the figure labeling

Reviewer #2 (Remarks to the Author): expertise in metabolism

This study showed that DJ-1 inhibits ferroptosis through preserving SAHH activity and promoting Hcy production. The study overall is novel and interesting, but there are quite several significant weaknesses that prevent its publication in the current form. The authors need to address the following concerns.

1. This study showed that DJ-1 knockdown affected ferroptosis induced by erastin and RSL3, but DJ-1 re-expression in DJ-1 KO MEFs only affected erastin but not RSL3-induced ferroptosis. How to explain this inconsistency?
2. Fig. 1C claimed that DJ-1 KD cells exhibit ferroptotic cell morphological changes under EM analysis (shrunken mitochondria with increased membrane density. Basically mitochondria appear to be smaller and darker). However, upon careful review of the data, it seems that, under erastin treatment condition, the mitochondria in ctrl KD cells are smaller and darker than those in DJ-1 KD cells, which is exactly opposite to their conclusion.
3. Fig. 5B showed that DJ-1 KD decreased Hcy level under basal condition (DMSO). Based on their model, decreased Hcy level should result in decreased cysteine levels and decreased GSH levels. However, Fig. 4A showed that DJ-1 KD did not affect GSH level under basal condition (only decreased GSH levels under erastin treatment). How to explain this? In addition, the authors should measure cysteine levels under the same experimental conditions.
4. Considering (1) the established the connection between DJ-1 and Nrf2, (2) and it is well established that SLC7A11 is a Nrf2 target and regulates ferroptosis, (3) the authors show that DJ1 affect GSH biosynthesis, it seems that the most straightforward hypothesis would be that DJ-1 regulates ferroptosis through SLC7A11. However, the authors completely ignored this possibility. They need to examine the effect of DJ-1 knockdown and re-expression on SLC7A11 expression and cystine uptake.
5. Figure 5 measured Hcy and SAH levels. The data from metabolite measurement sometime can be misleading. In order to substantiate their conclusion, the authors need to conduct flux analysis to measure transsulfuration flux (see Fig. 4 from PMID: 29346757).
6. Fig. 6C showed that DJ-1 overexpression attenuated whereas DJ-1 KD increased AHCYL1-SAHH interaction. However, the interaction increase is very minor. The authors need to quantify the data. In addition, they need to repeat the experiments to analyze the effects of DJ-1 overexpression or KD on endogenous AHCYL1-SAHH interaction (in the same cell lines in which they show DJ-1 KD or overexpression affects ferroptosis).
7. Based on their model, knockdown of AHCYL, a negative regulator of SAHH, should promote SAHH activity and Hcy production, thus inhibits ferroptosis. However, Fig. 6F shows that AHCYL KD did not affect lipid ROS upon erastin treatment. How to explain this?
8. Fig. 6F and Fig. 5N only show lipid ROS data. The authors also need to measure cell death in the same experiments.
9. In the current CRISPR era, it is a pity that the whole study never used CRISPR approach. The

key experiments need to be repeated by CRISPR approach (with at least two gRNAs).

10. In figure 7, the authors should also conduct 4HNE staining to measure lipid peroxidation in their tumor samples.

Reviewer #3 (Remarks to the Author): expertise in DJ-1

Cao et al. demonstrated that DJ-1 rescues cells from ferroptosis by preserving the enzymatic activity of SAHH and thereby keeping GSH. Their findings encourage the possibility of better cancer chemotherapy using ferroptosis inducers by silencing DJ-1. They showed various convincing data mostly in cultured cells.

Following issues should be considered and amended.

1) As for endocellular SAH and Hcy levels, the point to demonstrate is that Hcy level is lower, but upstream metabolite SAH is rather constant, in DJ-1 KD cells. The data shown in Fig. 5A look inconsistent with those in Fig. 5B and 5C for the cells treated with vehicle DMSO, which may puzzle the readers. The data in Fig. 5A are better to be deleted or move to supplement.

2) On lines 229-235, it is described that SAHH was consistent in terms of expression and posttranslational modifications either in DJ-1 KD or overexpressing cells, and that the decrease and increase of ectopic SAHH activity by KD and overexpression of DJ-1, respectively, were not reconstituted by use of recombinant proteins. The subsequent description 'We thus reasoned' on line 235 looks abrupt and rather illogical. Explain logically the reason to assume indirect effect of DJ-1 on the SAHH activity from the data above.

3) Overexpression of SAHH in DJ-1 KD cells decreased lipid ROS accumulation induced by Erastin and treatment of cells with SAH enhanced the decrease (Fig. 5M and 5N). This was founded on the results that the suppression of SAHH by RNA interference promoted Erastin-induced lipid ROS accumulation (Fig. S3E and S3F) and the data should be included in Fig. 5 prior to the current 5M and 5N, rather replacing the data of the SAH treatment (right half of 5N) which are not essential per se.

4) The result of BioID experiments identified AHCYL as a good candidate for DJ-1 target. Based on the data shown in Fig. 6C, further analyses were focused on AHCYL1. In Fig. 6C, all the signals in the right panel for AHCYL2 were weak, including input and GAPDH control, and it seems unreasonable to overlook the interaction with AHCYL2. Describe more the reasons to focus on AHCYL1, but not on AHCYL2, and/or background of the two proteins including functions or tissue-specific expression.

5) In Fig. 6D, 'Flag(s)' and 'Flag(l)', most probably representing short and long exposure, should be described in the legend.

6) In Fig. 7F, Hcy levels were compared between in DJ-1 KD tumors and in control KD tumors. The data were shown only for those treated with vehicle but not for those treated with PE. All the figures in Fig. 7 except 7F (7A-7E) included the data for both vehicle- and PE-treated tumors. The data for PE-treated tumors should also be included in Fig. 7F, otherwise it will be assumed the relevant data were intentionally not shown.

7) In Fig. 2, introduction of various mutants of DJ-1 into DJ-1^{-/-} MEF brought neither decrease of lipid ROS accumulation nor increase of viability after erastin treatment. Those effectless DJ-1 mutants (not all, but one or some) should be included in other reconstitution experiments as in Fig. D-E and H, overexpression introduction as in Fig. 5I-J. and the interaction assay with AHCYL-1

as in Fig. 6C.

Tipos

line 248, 'BiaID' should be 'BioID'

line 924, 'Figure S3' should be 'Figure S4'

We really appreciate the valuable comments and useful suggestions from the reviewers on our manuscript. We have taken all the comments into careful consideration and revised our manuscript accordingly. Please see the point-to-point response below.

Response to Reviewer #1

Comment 1. One major drawback of this work is that the extent of effects in many of the experiments appears to be modest or rather subtle. It remains unclear, at this point, that whether the lack of stronger metabolic and ferroptotic effects following DJ-1 knockdown is due to: 1) the weak knockdown efficacy of the two shRNAs that were used as the only means of DJ-1 suppression throughout the study; 2) the generally weak ferroptotic responses and incomplete death of the cell lines examined, which may have other ferroptosis-suppressive mechanisms that mask the effects of DJ-1; 3) the limited efficacy and specificity of Erastin in eliciting ferroptosis; 4) DJ-1's rather non-specific activity in diminishing cytosolic ROS; or 5) activation of compensatory pathways in response to DJ-1 suppression. To substantiate that DJ-1 does play a critical role in ferroptosis protection, the following alternative approaches are recommended: a. Include an alternative loss of function approach to inhibit DJ-1 deeply, such as CRISPR/Cas9-based knockout, and assess how ferroptosis and homocysteine metabolism is affected in those contexts. b. Present viability measurements of compound treated cells as wider range dose-response curves rather than selected concentrations. c. Use at least one alternative cell line (besides DJ-1^{-/-} MEF) that exhibits higher intrinsic sensitivity to ferroptosis to test whether exogenous DJ-1 expression protects the cells from ferroptosis, and whether such cell death can be rescued by ferrostatin or liproxstatin.

Response: Thank you for pointing this out. The key concern raised by Reviewer #1 is the reason for explaining the modest effect of DJ-1 knockdown on cell viability and the metabolic characterizations. We really appreciate the doable

approaches suggested by Reviewer #1. We have conducted all experiments mentioned by reviewer including (1) using CRISPR/Cas9-based knockout cells, (2) treating cells with a wider range of concentrations of compound, and (3) choosing the more sensitive cell lines. Additionally, we optimized the cell viability assay by increasing the compound treatment time. Based on these intensive studies, our results suggest that the modest effect is mainly attributed to two facts: one is the knockdown efficacy, and the other one is the cell viability assay condition.

Specifically, by utilizing CRISPR/Cas9-based knockout technology, we generated three DJ-1 knockout subclones from H1299 cells with two sgRNAs (new Fig.S1F and S1G). Subsequently, we tested the lipid ROS level, cell viability, and homocysteine level by using these KO cells. As shown in new Fig.1G, three DJ-1 KO subclones exhibited more accumulation of Erastin-induced lipid ROS comparing to three control subclones (wild type subclones). Indeed, comparing to shRNA-mediated knockdown system, CRISPR/Cas9-based knockout system provided greater effect size (lipid ROS % increased from 25.01% to 51.39% in knockdown system, and from 18.04% to 69.49% in knockout system). Similarly, we also observed that the effect size of cell viability in knockout system is greater than the one in knockdown system by treating cells with a wider range of concentrations of compound (new Fig.1D and 1H). Of note is the fact that the improper time point chosen in cell viability assay is another potential reason for the modest effect size of cell viability. When we prolong the treatment time (from 24h to 36h), the induction effect of ferroptotic cell death was dramatically enhanced in DJ-1 knockdown cells (Comparing new Fig.1F to old Fig.1F). Collectively, these data suggest that both knockdown efficacy and the assay condition are response for the modest effect size. Accordingly, we used these knockout cells and/or optimized cell viability assay to repeat the cell viability experiments with a wider range of concentrations of compound. Please refer to new Fig.1D, 1F, 1G, 1H, 3D, 4C, 4E, 5M, S1E, and S5D.

For the metabolic characterization part, we measured the level of endocellular GSH and homocysteine (Hcy) under ferroptotic condition using DJ-1 knockout subclones. As shown in new Fig.S3D and new Fig.5B, the DJ-1 knockout significantly decreased GSH levels and Hcy levels, and the effect size was stronger than the one in DJ-1 knockdown system (Fig.4A and Fig.5A). Thus, these data suggest that the modest effect size of metabolic characterization is mainly due to the insufficient suppression level of DJ-1.

The third approach proposed by Reviewer #1 is that choosing the higher sensitivity cell lines to test whether exogenous DJ-1 expression protects the cells from ferroptosis. As shown in new Fig.S8A and S8B, we chose nine cell lines including renal cell carcinoma and hepatocellular carcinoma, and found that 786-O, A2780 and KHOS are the most sensitive cell lines, whose sensitivities are similar to that of H1299. Then, we overexpressed DJ-1 in these cell lines and tested its protective effect on ferroptosis. However, we didn't observe any changes of lipid ROS accumulation and cell death caused by Erastin in these sensitive cell lines, which is in line with the results from H1299 (new Fig.S2A-S2E). We think these data suggest that the protective effect of overexpressed DJ-1 against ferroptosis is not a linear response, when the cells are expressing basal levels of DJ-1.

Fig.S1F**Fig.S1G****Fig.1G****Fig.1H****Fig.S3D****Fig.5B****Fig.4A****Fig.5A**
Fig.1D**Fig.1F****Fig.S1E****Fig.3D****Fig.4C****Fig.4E****Fig.5M****Fig.S5D**
Fig.S8A**Fig.S8B****Fig.S2A****Fig.S2B****Fig.S2C****Fig.S2E****Fig.S2D**
Newly released figures

Comment 2. While the authors started by speculating that DJ-1 may have a regulatory role in lipid ROS production and ferroptosis, the mechanistic dissection revealed that DJ-1 regulates glutathione synthesis through upregulating homocysteine production. Since glutathione is still a general antioxidant reagent that exhibits functions in and beyond ferroptosis, the manifestation of DJ-1's ferroptosis protective role could be highly context-dependent, i.e. stress-dependent. It is recommended that the authors go back to this point in the discussion, to clarify that DJ-1's anti-oxidative stress role is not specific for antagonizing lipid

hydroperoxides, but rather general for both cytosolic and lipid ROS. Along this line, the assessment of general ROS levels, e.g. by H₂-DCFDA in wildtype vs DJ-1 knockdown/knockout cells is recommended as a control.

Response: It's a good point. To further validate this point, we examined the general ROS level by H₂-DCFDA in control and DJ-1 knockdown cells after treating with Erastin as suggested by Reviewer #1. As shown in new Fig.S8C, the general ROS level was increased after Erastin treatment, and DJ-1 knockdown significantly enhanced this effect, suggesting manifestation of DJ-1's ferroptosis protective role is highly context-dependent. Taken together, our data demonstrated that DJ-1's anti-oxidative stress role is not specific for antagonizing lipid hydroperoxides, but rather general for both cytosolic and lipid ROS. We have brought this point into the discussion part accordingly.

Fig.S8C

Newly released Fig.S8C

Comment 3. (Optional) The *in vivo* experiment demonstrated a tumor suppressive effect of DJ-1 knockdown and piperazine erastin (PE) treatment, however, the connection of this effect to ferroptosis *in vivo* is only minimally justified by the usage of erastin, and the up-regulation of PTGS2, which can be induced by many cellular stress responses including inflammation, in addition to be upregulated in ferroptotic tissues. Hence a treatment arm with a lipophilic anti-oxidant that effectively and specifically blocks ferroptosis, such as liproxstatin (Friedmann Angeli et al., Nature Cell Biology 2014) is recommended for

such in vivo experiments.

Response: Thank you for reviewer's suggestion. The key point is that the connection of the tumor suppressive effect of DJ-1 knockdown to ferroptosis *in vivo* is only justified by the up-regulation of *PTGS2*. Accordingly, we determined the 4-HNE level, another classic marker of ferroptosis, in homogenized tumor tissues by applying the ELISA assay. As shown in new Fig.7F, knockdown of DJ-1 significantly increased the 4-HNE levels after PE treatment, which is consistent with the changes of *PTGS2*.

Since the reviewer mentioned that the up-regulation of *PTGS2* could be induced by inflammation, we subsequently examined the tumor tissue histology and inflammation status by Hematoxylin/eosin (H&E) and CD68 immunohistochemical staining respectively. As shown in new Fig.S8D, we didn't observe the increase of infiltration of inflammation associated cells, such as macrophages, in DJ-1 KD tissue treated with PE (cells in tumor tissue are mainly cancer cells). Similarly, the amount of CD68 positive staining cells (indicating macrophages) in tumor tissue was not changed among four groups, and much less than RAW264.7 and K7M2 mixed tumor sample from our previous report (*Cancer Immunol Res.* 2019 Feb;7(2):292-305.), which was set as the system control here (new Fig.S8D and S8E). Based on these results, we believe that the up-regulation of *PTGS2* is most probably due to the PE treatment effect.

Taken together, we conclude that DJ-1 silencing enhances the sensitivity of H1299 cells to PE-induced tumor growth arrest, showing that DJ-1 plays a critical role in ferroptosis *in vivo*. The figures and the manuscript were revised accordingly.

Fig.7F**Fig.7G****Fig.S8D****Fig.S8E**
Newly released figures

Comment 4. The authors showed in Figures 2 and S1D that RSL3-induced ferroptosis in DJ-1^{-/-} MEF cells was not reversed by DJ-1 overexpression, and suggested that DJ-1 does not directly affect GPX4. It seems unclear to this referee that if DJ-1 inhibition does have a substantial effect on glutathione abundance, how would the inhibition of GPX4 is completely unaffected by DJ-1 inhibition. It's important and relevant to clarify this point, perhaps by using a more specific GPX4 inhibitor, ML210 (Eaton et al., BioRxiv (doi: <https://doi.org/10.1101/376764>)), and by presenting the

viability results as wide range dose-response curves.

Response: If we understand this correctly, the reviewer's key question is why DJ-1 re-expression in DJ-1 KO MEFs only affected Erastin but not RSL3-induced ferroptosis. It's an interesting point. We performed the experiments suggested by Reviewer #1 using a more specific GPX4 inhibitor ML210 with presenting the viability results as wide range dose-response curves. As shown in new Fig.S2G, DJ-1 re-expression in DJ-1 KO MEFs was failed to diminish the lipid ROS and cell viability induced by ML210, which was in line with the results from RSL3-treated cells. These results imply that the results of GPX4 inhibitors in re-overexpression system are indeed existed and not a compound specific event.

We think the results from DJ-1 KO MEFs are consistent with our working model: **DJ-1 depletion can distinctly decreased Hcy levels, whereas it only remarkably inhibit the GSH level when the cystine import is blocked by SLC7A11 inhibitors (Erastin and Sorafenib).** As we illustrated in revised Fig.8, there are two routes to generate the intracellular cysteine in cells: one is the cystine import through SLC7A11 and the other one is the transsulfuration pathway. Under basal condition, the cystine import plays the dominate role for the cysteine generation. Our results suggest that the effect on SAHH by DJ-1 is a widespread fact. As our data shown in old Fig.5B (new Fig.5A), even without Erastin treatment, DJ-1 knockdown distinctly decreased Hcy levels, whereas there was no significant effect on GSH levels. This phenomenon might be due to the existence of cystine import through SLC7A11, though the transsulfuration pathway is affected. To confirm this, we measured intracellular cysteine levels in control and KD cells without Erastin treatment. As shown in new Fig.S3C, there is no difference in intracellular cysteine levels between control and DJ-1 KD cells, which is in line with our proposed model under basal condition. However, when the cystine import is blocked by Erastin, the cysteine generation might only rely on the transsulfuration pathway. That's why DJ-1 depletion can remarkably inhibit the GSH level only when the cystine import is

blocked by SLC7A11 inhibitors. In case of GPX4 inhibitors, it directly inhibits GPX4 and induced lipid ROS. This effect might not be affected by re-expression of upstream DJ1. This could be the reason to explain the DJ-1 re-expression in KO MEFs only reversed the effect of SLC7A11 inhibitors (Erastin and Sorafenib) but not GPX4 inhibitors (RSL3 and ML210). We have brought this point into the discussion part, and hope the results will be easier for the reader to understand.

Fig.S2G

Fig.5A

Fig.S3C

Newly released figures

Fig.8

Revised Fig.8

Comment 5. Grammatical issues.

Response: We apologize for the grammatical issues. Our revised manuscript has been edited by two experts of language editing from *American Journal Experts*. All the changes are highlighted in the revised manuscript.

Response to Reviewer #2

Comment 1. This study showed that DJ-1 knockdown affected ferroptosis induced by erastin and RSL3, but DJ-1 re-expression in DJ-1 KO MEFs only affected erastin but not RSL3-induced ferroptosis. How to explain this inconsistency?

Response: If we understand this correctly, the reviewer's key question is why DJ-1 re-expression in DJ-1 KO MEFs only affected Erastin but not RSL3-induced ferroptosis. It's an interesting point and Reviewer #1 think it might be due to the low specificity of RSL3 on GPX4. Thus, we performed the

experiments suggested by Reviewer #1 using a more specific GPX4 inhibitor ML210 with presenting the viability results as wide range dose-response curves. As shown in new Fig.S2G, DJ-1 re-expression in DJ-1 KO MEFs was failed to diminish the lipid ROS and cell viability induced by ML210 (new Fig.S2G), which was in line with the results from RSL3-treated cells. These results imply that the results of GPX4 inhibitors in re-overexpression system is indeed existed and not a compound specific event.

We think the results from DJ-1 KO MEFs is consistent with our working model: DJ-1 depletion can distinctly decreased Hcy levels, whereas it only remarkably inhibit the GSH level when the cystine import is blocked by SLC7A11 inhibitors (Erastin and Sorafenib). As we illustrated in revised Fig.8, there are two routes to generate the intracellular cysteine in cells: one is the cystine import through SLC7A11 and the other one is the transsulfuration pathway. Under basal condition, the cystine import plays the dominate role for the cysteine generation. Our results suggest that the effect on SAHH by DJ-1 is a widespread fact. As our data shown in old Fig.5B (new Fig.5A), even without Erastin treatment, DJ-1 knockdown distinctly decreased Hcy levels, whereas there was no significant effect on GSH levels. This phenomenon might be due to the existence of cystine import through SLC7A11, though the transsulfuration pathway is affected. That's why DJ-1 depletion can remarkably inhibit the GSH level only when the cystine import is blocked by SLC7A11 inhibitors. In case of GPX4 inhibitors, it directly inhibits GPX4 and induced lipid ROS. This effect might not be affected by re-expression of upstream DJ1. This could be the reason to explain the DJ-1 re-expression in KO MEFs only reversed the effect of SLC7A11 inhibitors (Erastin and Sorafenib) but not GPX4 inhibitors (RSL3 and ML210). We have brought this point into the discussion part, and hope the results will be easier for the reader to understand.

Fig.S2G

Fig.5A

Fig.S3C

Newly released figures

Fig.8

Revised Fig.8

Comment 2. Fig. 1C claimed that DJ-1 KD cells exhibit ferroptotic cell morphological changes under EM analysis (shrunken mitochondria with

increased membrane density. Basically mitochondria appear to be smaller and darker). However, upon careful review of the data, it seems that, under erastin treatment condition, the mitochondria in ctrl KD cells are smaller and darker than those in DJ-1 KD cells, which is exactly opposite to their conclusion.

Response: We sincerely apologize for the confusion about the cell morphological changes. The description is correct, however, we accidentally mess up the images in old Fig.1C. After checking back the raw data, we found that the image of DJ-1 KD cells treated with Erastin (the swelling mitochondrial) is actually the DJ-1 KD cells cultured under hypoxia for 16h (from another unrelated project we were working with). To make sure DJ-1 KD cells do exhibit ferroptotic cell morphological changes, we repeated this experiment with two KD cells. As shown in new Fig.1C, we did consistently observe shrunken mitochondria with increased membrane density in two DJ-1 KD cells treated with Erastin. Together with lipid ROS and cell viability assay results, DJ-1 plays a critical role in ferroptotic cell death, Again, we appreciate the review's kind remind, and really feel sorry for this serious mistake we have made. We further carefully checked the data throughout the manuscript and guarantee that there is no such kind of mistake existed in the revised paper.

Fig.1C

Newly released Fig.1C

Comment 3. Fig. 5B showed that DJ-1 KD decreased Hcy level under basal condition (DMSO). Based on their model, decreased Hcy level should result in decreased cysteine levels and decreased GSH levels. However, Fig. 4A showed that DJ-1 KD did not affect GSH level under basal condition (only decreased GSH levels under erastin treatment). How to explain this? In addition, the authors should measure cysteine levels under the same experimental conditions.

Response: Thank you for the suggestion. As we illustrated in revised Fig.8, cysteine is the precursor for GSH synthesis. There are two routes to generate the intracellular cysteine in cells: one is the cysteine import through SLC7A11 and the other one is the transsulfuration pathway. Under basal condition, the cysteine import plays the dominant role for the cysteine generation, while the transsulfuration pathway plays an alternative role. Our results suggest that the

effect on SAHH by DJ-1 is a widespread fact. As our data shown in old Fig.5B (new Fig.5A), even without Erastin treatment, DJ-1 knockdown distinctly decreased Hcy levels, whereas there was no significant effect on GSH levels. This phenomenon might be due to the existence of cystine import though SLC7A11, though the transsulfuration pathway is affected.

To confirm this, we measured intracellular cysteine levels in control and KD cells without Erastin treatment as suggested by Reviewer #2. As shown in new Fig.S3C, there is no difference in intracellular cysteine levels between control and DJ-1 KD cells, which is in line with our proposed model under basal condition. Therefore, under basal condition, DJ-1 knockdown distinctly decreased Hcy levels, whereas there was no significant effect on cysteine and GSH levels. However, when the cystine import is blocked, the cysteine generation might only rely on the transsulfuration pathway. That's why DJ-1 depletion can remarkably inhibit the GSH level only when the cystine import is blocked. We realize that the confusion might be caused by the unclear way we presented in old Fig.8. So in the revised manuscript, we modified the Fig.8 accordingly.

Newly released figures

Fig.8

Revised Fig.8

Comment 4. Considering (1) the established the connection between DJ-1 and Nrf2, (2) and it is well established that SLC7A11 is a Nrf2 target and regulates ferroptosis, (3) the authors show that DJ1 affect GSH biosynthesis, it seems that the most straightforward hypothesis would be that DJ-1 regulates ferroptosis through SLC7A11. However, the authors completely ignored this possibility. They need to examine the effect of DJ-1 knockdown and re-expression on SLC7A11 expression and cystine uptake.

Response: It's a good point. We have examined the effect of DJ-1 knockdown and re-expression on the expression level of SLC7A11. As shown in new Fig.S3A, the SLC7A11 protein level was not significantly changed in DJ-1 knockdown H1299 cells. Moreover, under the Erastin treatment condition, Nrf2-mediated increase of SLC7A11 was also not diminished upon DJ-1 knockdown. For the function of SLC7A11, we have examined the alteration of cystine uptake upon DJ-1 knockdown. As shown in new Fig.S3B, there was no difference in

cystine uptake in DJ-1 knockdown cells, indicating the function of SLC7A11 was not affected by DJ-1 knockdown.

Another important fact should be mentioned is that if DJ-1 regulates ferroptosis through SLC7A11, the DJ-1 knockdown itself should dramatically trigger the ferroptotic cell death. However, our data in Fig.1 imply that the DJ-1 knockdown itself is not necessary to induce ferroptosis. Taken together, these results clearly rule out the possibility that DJ-1 regulates ferroptosis through SLC7A11. We sincerely appreciate the reviewer's valuable comments. We have added these results and description into our revised manuscript accordingly.

Newly released Fig.S3A and S3B

Comment 5. Figure 5 measured Hcy and SAH levels. The data from metabolite measurement sometime can be misleading. In order to substantiate their conclusion, the authors need to conduct flux analysis to measure transsulfuration flux.

Response: Thank you for reviewer's suggestion. As suggested, we investigated whether DJ-1 knockdown disrupt the generation of Hcy by limiting transsulfuration pathway using mass spectrometry and carbon-13 labeling. H1299 control and DJ-1 knockdown cells were pretreated with methionine starvation and then incubated for 4 h in EBSS supplemented with methionine-1-¹³C. As shown in new Fig.S4B, depletion of DJ-1 resulted in lower levels of total Hcy (all isotopologues) (0.64 fold compared to control cells). Moreover, we observed that transsulfuration pathway was inhibited in DJ-1 knockdown cells by a reduction in Hcy M+1 relative to control cells (new Fig.S4C). Taken

together, these results suggest that DJ-1 knockdown decreased the Hcy level. The figures and the manuscript were revised accordingly, and hope it will be better for readers to understand.

Fig.S4A

Fig.S4B

Fig.S4C

Comment 6. Fig. 6C showed that DJ-1 overexpression attenuated whereas DJ-1 KD increased AHCYL1-SAHH interaction. However, the interaction increase is very minor. The authors need to quantify the data. In addition, they need to repeat the experiments to analyze the effects of DJ-1 overexpression or KD on endogenous AHCYL1-SAHH interaction (in the same cell lines in which they show DJ-1 KD or overexpression affects ferroptosis).

Response: Thank you for reviewer's suggestion. As suggested, we have quantified the AHCYL1-SAHH interaction data from three independent experiments. As shown in new Fig.S6A, DJ-1 overexpression attenuated whereas DJ-1 KD increased AHCYL1-SAHH interaction. Though the interaction increase was minor, the increase was consistently observed and there was a statistical difference. To better draw our conclusion, we analyzed the effects of

DJ-1 KD on endogenous AHCYL1-SAHH interaction in H1299 cells as suggested by Reviewer #2. Again, the DJ-1 KD consistently increased endogenous AHCYL1-SAHH interaction (new Fig.S6B). We have added these results and description into our revised manuscript accordingly.

Fig.S6A

Fig.S6B

Newly released Fig.S6A and S6B

Comment 7. Based on their model, knockdown of AHCYL, a negative regulator of SAHH, should promote SAHH activity and Hcy production, thus inhibits ferroptosis. However, Fig. 6F shows that AHCYL KD did not affect lipid ROS upon earstin treatment. How to explain this?

Response: It's an interesting point and we were also confused with this result. Fortunately, the comments from Reviewer #1 remind us that this inconsistency could be due to the selected concentration treatment, especially the effect of 2 μ M Erastin on ferroptosis induction is modest. Thus, we repeated this experiment with a higher concentration of Erastin. As shown in new Fig.S8F, we found that AHCYL1 KD affect lipid ROS upon 4 μ M Erastin treatment as expected. Collectively, these data, together with other evidence, further confirmed that the regulation of SAHH activity by DJ-1 was mediated by AHCYL1.

Fig.S8F**Fig.S8G**
Newly released Fig.S8F and S8G

Comment 8. Fig. 6F and Fig. 5N only show lipid ROS data. The authors also need to measure cell death in the same experiments.

Response: Thank you for pointing this out. We have performed the cell viability assay as suggested. As shown in revised Fig.5M and S5D, overexpressed SAHH could reverse the increase of ferroptosis caused by DJ-1 knockdown. Moreover, AHCYL1 knockdown reverses the increase of ferroptosis caused by DJ-1 knockdown with a wider range of concentrations of Erastin (new Fig.S6D and new S8G). The figures and the context are revised accordingly.

Fig.5M**Fig.S5D****Fig.S6D****Fig.S8G**
Newly released figures

Comment 9. In the current CRISPR era, it is a pity that the whole study never used CRISPR approach. The key experiments need to be repeated

by CRISPR approach (with at least two gRNAs).

Response: Thank you for reviewer's suggestion. By utilizing CRISPR/Cas9-based knockout technology, we generated three DJ-1 knockout subclones from H1299 cells with two sgRNAs (new Fig.S1F and S1G). Subsequently, we evaluated the lipid ROS level, cell viability, GSH level and homocysteine level by using these DJ-1 knockout cells. As shown in new Fig.1G, three DJ-1 KO subclones exhibited more Erastin-induced accumulation of lipid ROS comparing to three WT subclones. Similarly, we also observed that more ferroptotic cell death was induced in DJ-1 knockout subclones comparing to wild type subclones (new Fig.1H). For the metabolic characterization part, we measured the level of endocellular GSH and homocysteine (Hcy) under ferroptotic condition using DJ-1 knockout subclones. As shown in new Fig.S3D and new Fig.5B, the DJ-1 knockout significantly decreased GSH levels and Hcy levels. Thus, these data from CRISPR system further strengthen our conclusions. The figures and the context are revised accordingly.

Fig.S1F

PARK7 exon2

DJ-1 WT GCTTGT.....ATGGCTTCCAAAAGAG.....CTGTAGATGTC.....GCTGGG⁹⁰

DJ-1 KO#1 GCTTGT.....ATGGCTTCCAAAGAG.....CTGTAGATGTC.....GCTGGG

DJ-1 KO#2 GCTTGT.....ATGGCTTCCAAATGAG.....CTGTAGATGTC.....GCTGGG

Fig.S1G

PARK7 exon3

DJ-1 WT GCTGGG⁹¹ATTAAGGTC.....AAAAAAGAG¹⁹²GGACCA

DJ-1 KO#3 GCTGGGGGACCA

H1299

Fig.1G**Fig.1H****Fig.S3D****Fig.5B**
Newly released figures

Comment 10. In figure 7, the authors should also conduct 4HNE staining to measure lipid peroxidation in their tumor samples.

Response: Thank you for reviewer's suggestion. We determined the 4-HNE level in homogenized tumor tissues by applying the ELISA assay. As shown in new Fig.7F, knockdown of DJ-1 significantly increased the 4-HNE level after PE treatment, which is consistent with the changes of *PTGS2*. Based on these results, we concluded that DJ-1 silencing enhances the sensitivity of H1299 cells to PE-induced tumor growth arrest, showing that DJ-1 plays a critical role in ferroptosis *in vivo*. The figures and the manuscript are revised accordingly.

Newly released Fig.7E and 7F

Response to Reviewer #3

Comment 1. As for endocellular SAH and Hcy levels, the point to demonstrate is that Hcy level is lower, but upstream metabolite SAH is rather constant, in DJ-1 KD cells. The data shown in Fig. 5A look inconsistent with those in Fig. 5B and 5C for the cells treated with vehicle DMSO, which may puzzle the readers. The data in Fig. 5A are better to be deleted or move to supplement.

Response: Thank you for reviewer's suggestion. We have conducted flux analysis as suggested by reviewer #2. We investigated whether DJ-1 knockdown disrupt the generation of Hcy by limiting transsulfuration pathway using mass spectrometry and carbon-13 labeling. H1299 control and DJ-1 knockdown cells were pretreated with methionine starvation and then incubated for 4 h in EBSS supplemented with methionine-1-¹³C. As shown in new Fig.S4B, depletion of DJ-1 resulted in lower levels of total Hcy (all isotopologues) (0.64 fold compared to control cells). Moreover, we observed that transsulfuration pathway was inhibited in DJ-1 knockdown cells by a reduction in Hcy M+1

relative to control cells (new Fig.S4C). Taken together, these results suggest that DJ-1 knockdown decreased the Hcy level. We also moved the results to supplement as suggested by Reviewer #3. Please refer to new Fig.S4A-S4C.

Newly released Fig.S4A-S4C

Comment 2. On lines 229-235, it is described that SAHH was consistent in terms of expression and posttranslational modifications either in DJ-1 KD or overexpressing cells, and that the decrease and increase of ectopic SAHH activity by KD and overexpression of DJ-1, respectively, were not reconstituted by use of recombinant proteins. The subsequent description ‘We thus reasoned’ on line 235 looks abrupt and rather illogical. Explain logically the reason to assume indirect effect of DJ-1 on the SAHH activity from the data above.

Response: Thank you for pointing this out. To understand how DJ-1 affects the activity of SAHH, we first hypothesized the direct effect of DJ-1 SAHH activity. To test this, we purified recombinant DJ-1 and SAHH *in vitro* and subsequently determined the SAHH activity. However, unlike the intracellular SAHH activity results shown in revised Fig.5H, recombinant DJ-1 didn't enhance the SAHH activity by using this reconstitute assay (revised Fig. S5E). Thus, we think DJ-1 affect SAHH activity indirectly. Then, we tested whether DJ-1 influences the

expression and post-translational modification of SAHH. As shown in revised Fig.S5F-S5I, DJ-1 did not change the protein level of SAHH as well as its reported post-translational modifications such as acetylation and phosphorylation level. We thus reasoned that DJ-1 might indirectly affect the activity of SAHH through another partner protein. To test this, we used the BiOD assay to identify the AHCYL in further analysis. We have rearranged the context in a more logical way and hope it will be better for readers to understand.

Fig.5H

Fig.S5E

Fig.S5F

Fig.S5G

Fig.S5H

Fig.S5I

Re-organized figures

Comment 3. Overexpression of SAHH in DJ-1 KD cells decreased lipid ROS accumulation induced by Erastin and treatment of cells with SAH enhanced the decrease (Fig. 5M and 5N). This was founded on the results that the suppression of SAHH by RNA interference promoted Erastin-induced lipid ROS accumulation (Fig. S3E and S3F) and the data should be included in Fig, 5 prior to the current 5M and 5N, rather replacing the data of the SAH treatment (right half of 5N) which are not essential per se.

Response: Thank you for reviewer's suggestion. We have reorganized our

figures according to Reviewer #3's suggestion. Please refer to re-organized Fig.5I, 5J, S5C, and S5D.

Fig.5I

Fig.5J

Fig.S5C

Fig.S5D

Re-organized figures

Comment 4. The result of BiOLD experiments identified AHCYL as a good candidate for DJ-1 target. Based on the data shown in Fig. 6C, further analyses were focused on AHCYL1. In Fig. 6C, all the signals in the right panel for AHCYL2 were weak, including input and GAPDH control, and it seems unreasonable to overlook the interaction with AHCYL2. Describe more the reasons to focus on AHCYL1, but not on AHCYL2, and/or background of the two proteins including functions or tissue-specific expression.

Response: It's a good point. The key concern raised by Reviewer #3 is the reason for choosing AHCYL1 for further study. In the previous version of manuscript, we reasoned that some background signal was detected in AHCYL2 groups, suggesting that AHCYL1 was relatively specific. In current revised manuscript, we provide more experimental evidences for choosing AHCYL1 for further analysis. As shown in new Fig.S6E-S6G, unlike AHCYL1

knockdown, AHCYL2 knockdown did not affect lipid ROS and cell viability in DJ-1 knockdown cells upon Erastin treatment. Moreover, DJ-1 overexpression and knockdown didn't affect the AHCYL2-SAHH interaction (new Fig.S6C). Collectively, these results further provide the new reasons for choosing AHCYL1 for further study. The figures and the manuscript were revised accordingly.

Fig.S6E

Fig.S6F

Fig.S6G

Fig.S6C

Newly released figures

Comment 5. In Fig. 6D, 'Flag(s)' and 'Flag(l)', most probably representing short and long exposure, should be described in the legend.

Response: The reviewer is right. 's' and 'l' represent short and long exposure respectively. We apologize for missing this information in the figure legend. In the revised manuscript, we have added these information accordingly. Thank you for the remind.

Comment 6. In Fig. 7F, Hcy levels were compared between in DJ-1 KD tumors and in control KD tumors. The data were shown only for those treated with vehicle but not for those treated with PE. All the figures in Fig.

7 except 7F (7A-7E) included the data for both vehicle- and PE-treated tumors. The data for PE-treated tumors should also be included in Fig. 7F, otherwise it will be assumed the relevant data were intentionally not shown.

Response: Thank you for reviewer's suggestion. We have examined the Hcy level in PE-treated tumors. As shown in new Fig.7G, Hcy levels were significantly lower in DJ-1 knockdown tumors compared with control group under both vehicle and PE treatment condition, which is in line with our results in cells (Fig.5).

Newly released Fig.7G

Comment 7. In Fig.2, introduction of various mutants of DJ-1 into DJ-1^{-/-}MEF brought neither decrease of lipid ROS accumulation nor increase of viability after erastin treatment. Those effectless DJ-1 mutants (not all, but one or some) should be included in other reconstitution experiments as in Fig. 5D-E and H, overexpression introduction as in Fig. 5I-J. and the interaction assay with AHCYL-1 as in Fig. 6C.

Response: It's another good point. We chose two DJ-1 mutants M26I and A104T to perform the experiments suggested by Reviewer #3. As shown in new Fig.S4F, a significant increase in Hcy levels, but not SAH levels, was observed in wild type but not two mutant DJ-1 re-expressed DJ-1^{-/-}MEFs. The similar results were also observed using the metabolic tracing system we established. As shown in new Fig.S4G, a significant increase in the Hcy levels was observed

in wild type but not two mutant DJ-1 re-expressed DJ-1^{-/-} MEFs. As shown in new Fig.S4I, the enzymatic activity of SAHH was dramatically increased by co-expressing wild type but not two mutant DJ-1. In addition, we also performed the interaction assay with AHCYL1 by using the WT and two mutants. As shown in new Fig.S6H, AHCYL1 were only labeled with biotin resulted from interaction with wild type BirA*-DJ-1, however no signal was detected in two mutants BirA*-DJ-1. Taken together, these results clearly confirmed that the proper functioning of DJ-1 is required for its preserving role in SAHH activity.

Fig.S4F

Fig.S4G

Fig.S4I

Fig.S6H

Newly released figures

Reviewers' comments:

Reviewer #1 (Remarks to the Author):

The authors have properly addressed all the concerns of this referee.

Several grammatical changes are recommended:

Page 5, line 101, "will markedly amplified" should be "markedly amplified"

Page 7, line 126, "modular" should be "modulator"

Page 13, line 263, "based the score" should be "based on the scores"

Page 13, line 273, "resulted" should be "result"

Page 14, line 284, "of important", should be "Importantly"

Page 15, line 307, "T/C value" should be defined

Reviewer #2 (Remarks to the Author):

The authors addressed most of the questions raised by this reviewer and have improved their manuscript. However, as detailed below, based on their updated model proposed in Fig. 8, there is a major conceptual issue to explain their loss-of-function data.

In the previous review, this reviewer asked the question "This study showed that DJ-1 knockdown affected ferroptosis induced by erastin and RSL3, but DJ-1 re-expression in DJ-1 KO MEFs only affected erastin but not RSL3-induced ferroptosis. How to explain this inconsistency?" Here this reviewer asked the authors to explain why their data from different cell lines/approaches are inconsistent. The authors responded by proposing a modified model to explain why DJ-1 specifically regulates erastin but not RSL3-induced ferroptosis. This model would make sense if all their data support this model. However, all their loss-of-function data (with shRNA and CRISPR KO) are in conflict with this model.

Specifically, since they show that DJ-1 KD/KO does not affect intracellular cysteine and GSH levels under BASAL conditions (Fig. 4A, Fig. S3D-3C, note that DJ-1 KD only decreased GSH levels under erastin treatment condition, presumably because cystine uptake provides the major support for intracellular cysteine, as proposed in Fig. 8), how would DJ-1 KD/KO affect sensitivity to GPX4 inhibition under these basal conditions? (when they compared cellular sensitivity in DJ-1 WT and KD/KO cells to GPX4 inhibitors, they did not block cystine uptake in those cells, therefore intracellular cysteine and GSH levels should be the same between DJ-1 WT and KD/KO cells. If so, what causes the sensitivity difference to GPX4 inhibitors between DJ-1 WT and KD/KO?)

Fig. 1C: it's still unclear to this reviewer how the EM data (under Erastin treatment) shows more "shrunken mitochondria with increased membrane density" in DJ-1 KD cells than control cells. Of note, there are more orange triangles pointing to mitochondria in DJ-1 KD cells than in control cells. However, in control cell + erastin image, there are several un-highlighted mitochondria which appear to be even smaller than the highlighted one.

Reviewer #3 (Remarks to the Author):

The authors verified well the points raised by the reviewers and showed more convincing data of additional experiments. The revised manuscript has been improved in a satisfactory manner by reorganizing and clarifying the description.

We really appreciate the valuable comments from the reviewers on our manuscript. We have taken all the comments into careful consideration and revised our manuscript accordingly. Please see the point-to-point response below.

Response to Reviewer #1

Comment 1. Several grammatical changes are recommended.

Response: Thank you for pointing these out. We have revised the manuscript accordingly as suggested.

Response to Reviewer #2

Comment 1. Here this reviewer asked the authors to explain why their data from different cell lines/approaches are inconsistent. The authors responded by proposing a modified model to explain why DJ-1 specifically regulates erastin but not RSL3-induced ferroptosis. This model would make sense if all their data support this model. However, all their loss-of-function data (with shRNA and CRISPR KO) are in conflict with this model. Specifically, since they show that DJ-1 KD/KO does not affect intracellular cysteine and GSH levels under BASAL conditions (Fig. 4A, Fig. S3D-3C, note that DJ-1 KD only decreased GSH levels under erastin treatment condition, presumably because cystine uptake provides the major support for intracellular cysteine, as proposed in Fig. 8), how would DJ-1 KD/KO affect sensitivity to GPX4 inhibition under these basal conditions?

Response: It's an interesting point. First, we think the data from different cell lines/approaches, especially for the Erastin data, is highly consistent. As the data shown in Fig.1 and Fig.2, DJ-1 KD/KO enhanced the Erastin-induced ferroptosis and DJ-1 re-expression compromised the Erastin's effect in DJ-1

KO cells. Thus, the conclusion is appropriated basing on our Erastin's data

The key issue is why DJ-1 KD/KO affect sensitivity to GPX4 inhibition? Because if there is no change of GSH level, it will not affect the effect of GPX4 inhibitors. And all this assumption is based on the limited knowledge of ferroptosis: GPX4, function as the only known ferroptosis suppressor, consumes GSH to scavenge lipid ROS. However, the knowledge of this field is expanding rapidly. At most recently, two back-to-back elegant papers published in *Nature* demonstrated that FSP1 functioned as a glutathione-independent ferroptosis suppressor (*Nature* (2019) doi:10.1038/s41586-019-1707-0; *Nature* (2019) doi:10.1038/s41586-019-1705-2), suggesting there could be more suppressors and regulatory ways to coordinately dictate ferroptosis. Thus, in DJ-1 KD cells, even if there is no change of GSH level, it might also affect the effect of GPX4 inhibitors through other mechanisms. Given that (1) the major finding of current study is deciphering DJ-1-dictated SAHH and its related transsulfuration pathway in ferroptosis (the upstream of the ferroptosis), (2) actually we mainly used Erastin for our study and only draw the Erastin in our scheme, the deep discovering how DJ-1 affect GPX4 inhibitors' effect might be out of scope in our paper, which is also an interesting puzzle in the ferroptosis field. In order to address this issue, we now bring these points into our discussion part so the readers wouldn't be confused or mis-led. We hope the reviewer will be satisfied with our solution for this concern.

Comment 2. It's still unclear to this reviewer how the EM data (under Erastin treatment) shows more "shrunken mitochondria with increased membrane density" in DJ-1 KD cells than control cells. Of note, there are more orange triangles pointing to mitochondria in DJ-1 KD cells than in control cells. However, in control cell + erastin image, there are several un-highlighted mitochondria which appear to be even smaller than the

highlighted one.

Response: Thank you for pointing this out. For this concern, it might be caused by the way we presented the data. As you can see in Fig. 1C, we used the orange triangles to indicate the classic morphologic changes of mitochondrial with shrunken mitochondria with increased membrane density. We agree with the fact that there are “several un-highlighted mitochondria which appear to be even smaller than the highlighted one”. However, there are even more these smaller organelles shown in DJ-1 KD cells, especially under the Erastin treatment condition (please refer to Fig.1C, DJ-1 KD groups). Since these smaller organelles are hard to be characterized as mitochondrial, we only highlighted the ones with clear mitochondrial structure in the figures. We also provide some other representative images below. Generally speaking, more shrunken mitochondria with increased membrane density were observed in DJ-1 KD cells treated with Erastin. As suggested by editor, we revised the Fig.1C with a more representative images accordingly.

EM data are shown below:

REVIEWERS' COMMENTS:

Reviewer #2 (Remarks to the Author):

The authors have adequately addressed the remaining concerns from this reviewer. The paper can be accepted.

Response letter

Response to Reviewer #2

Comment. The authors have adequately addressed the remaining concerns from this reviewer. The paper can be accepted..

Response: We really appreciate all the useful comments and suggestions from this reviewer.